# Beyond Trivial Counterfactual Generations with Diverse Valuable Explanations

## Abstract

Explainability of machine learning models has gained considerable attention within our research community given the importance of deploying more reliable machine-learning systems. Explanability can also be helpful for model debugging. In computer vision applications, most methods explain models by displaying the regions in the input image that they focus on for their prediction, but it is difficult to improve models based on these explanations since they do not indicate why the model fail. Counterfactual methods, on the other hand, indicate how to perturb the input to change the model prediction, providing details about the model's decision-making. Unfortunately, current counterfactual methods make ambiguous interpretations as they combine multiple biases of the model and the data in a single counterfactual interpretation of the model's decision. Moreover, these methods tend to generate trivial counterfactuals about the model's decision, as they often suggest to exaggerate or remove the presence of the attribute being classified. Trivial counterfactuals are usually not valuable, since the information they provide is often already known to the system's designer. In this work, we propose a counterfactual method that learns a perturbation in a disentangled latent space that is constrained using a diversity-enforcing loss to uncover multiple valuable explanations about the model's prediction. Further, we introduce a mechanism to prevent the model from producing trivial explanations. Experiments on CelebA and Synbols demonstrate that our model improves the success rate of producing high-quality valuable explanations when compared to previous state-of-the-art methods. We will make the code public.

## 1 Introduction

Consider a face authentication system for unlocking a device. In case of non-authentications (possible false-negative predictions), this system could provide generic advices to its user such as "face the camera" or "remove any face occlusions". However, these may not explain the reason for the possible malfunction. To provide more insights regarding its decisions, the system could instead provide information specific to the captured image (its input data). It might list the input features that most contributed to its decision (*e.g.*, a region of the input image), but this feature could be "face", which is trivial and does not suggest an alternative action to its user. Further, it provides little useful information about the model. Instead, non-trivial explanations may be key for better understanding and diagnosing the system— including the data it was trained on— and improving its reliability. Such explanations might improve systems across a wide variety of domains including in medical imaging [58], automated driving systems [48], and quality control in manufacturing [22].

The explainability literature aims to understand the decisions made by a machine learning (ML) model such as the aforementionned face authentication system. Counterfactual explanation methods [11, 13, 4] can help discover the limitations of a ML model by uncovering data and model biases. The counterfactual explanation methods provide perturbed versions of the input data that emphasize features that contributed most to the ML model's output. For example, if an authentication system is not recognizing a user wearing sunglasses then the system could generate an alternative image of the user's face without sunglasses that would be correctly recognized. This is different from other types of explainability methods such as feature importance methods [50, 51, 4] and boundary approximation methods [47, 37]. The former highlight salient regions of the input but do not indicate how the ML model could achieve a different prediction. The second family of methods produce

explanations that are limited to linear approximations of the ML model. Unfortunately, these linear approximations are often inaccurate. In contrast, counterfactual methods suggest changes in the input that would lead to a change in the corresponding output, providing information not only about where the change should be but also *what* the change should be.

Counterfactual explanations should be actionable, *i.e.*, a user should be able to act on it. An actionable explanation would suggest feasible changes like removing sunglasses instead of unrealistic ones like adding more eyes to the user's face. Counterfactual explanations that are *valid*, *proximal*, and *sparse* are more likely to be actionable [49, 38]. That is, a counterfactual explanation that changes the outcome of the ML model (*valid*) by changing the minimal number of input features (*sparse*), while remaining close to the input (*proximal*). Generating a set of *diverse* explanations increases the likelihood of finding an actionable explanation [49, 38]. A set of counterfactuals is diverse if each one proposes to change a different set of attributes. Intuitively, each of these explanations shed light on a different action that user can take to change the ML model's outcome.

Current counterfactual generation methods like xGEM [26] generates a single explanation that is far from the input. Thus, it fails to be *proximal*, *sparse*, and *diverse*. Progressive Exaggeration (PE) [53] provides higher-quality explanations more *proximal* than xGEM, but it still fails to provide a *diverse* set of explanations. In addition, the image generator of PE is trained on the same data as the ML model in order to detect biases thereby limiting their applicability. Moreover, like the previous methods in the literature, these two methods tend to produce *trivial* explanations. For instance, an explanation that suggests to increase the 'smile' attribute of a 'smile' classifier for an already-smiling face is trivial and it does not explain why a misclassification occurred. In this work, we focus on *diverse valuable* explanations (DiVE), that is, *valid*, *proximal*, *sparse*, and *non-trivial*.

We propose Diverse Valuable Explanations (DiVE), an explainability method that can interpret a ML model by identifying sets of valuable attributes that have the most effect on the ML model's output. DiVE produces multiple counterfactual explanations which are enforced to be *valuable*, and *diverse* resulting in more *actionable* explanations than the previous literature. Our method first learns a generative model of the data using a $\beta$-TCVAE [5] to obtain a disentangled latent representation which leads to more *proximal* and *sparse* explanations. In addition, the VAE is not required to be trained on the same dataset as the ML model to be explained. DiVE then learns a latent perturbation using constraints to enforce *diversity*, *sparsity*, and *proximity*. In order to generate *non-trivial* explanations, DiVE leverages the Fisher information matrix of its latent space to focus its search on the less influential factors of variation of the ML model. This mechanism enables the discovery of spurious correlations learned by the ML model.

We provide experiments to assess whether our explanations are more *valuable* and *diverse* than current state-of-the-art. First, we assess their *validity* on the CelebA dataset [33] and provide quantitative and qualitative results on a bias detection benchmark [53]. Second, we show that the generated explanations are more *proximal* in terms of Fréchet Inception Distance (FID) [19], which is a measure of similarity between two datasets of images commonly used to evaluate the generation quality of GAN. In addition, we evaluate the latent space closeness and face verification accuracy, as reported by Singla et al. [53]. Third, we assess the *sparsity* of the generated counterfactuals by computing the average change in facial attributes. Fourth, we show that DiVE is more successful at finding more *non-trivial* explanations than previous methods and baselines. In the supplementary material we provide additional results on the out-of-distribution performance of DiVE.

We summarize the contributions of this work as follows: 1) We propose DiVE, an explainability method that can interpret a ML model by identifying the attributes that have the most effect on its output. 2) DiVE achieves state of the art in terms of the *validity*, *proximity*, and *sparsity* of its explanations, detecting biases on the datasets, and producing multiple explanations for an image. 3) We identify the importance of finding *non-trivial* explanations and we propose a new benchmark to evaluate how *valuable* the explanations are. 4) We propose to leverage the Fisher information matrix of the latent space for finding spurious features that produce *non-trivial* explanations.

## 2 RELATED WORK

Explainable artificial intelligence (XAI) is a suite of techniques developed to make either the construction or interpretation of model decisions more accessible and meaningful. Broadly speaking, there are two branches of work in XAI, ad-hoc and post-hoc. Ad-hoc methods focus on making mod-

els interpretable, by imbuing model components or parameters with interpretations that are rooted in the data themselves [45, 39, 25]. Unfortunately, most successful machine learning methods, including deep learning ones, are uninterpretable [6, 32, 18, 24].

Post-hoc methods aim to explain the decisions of non interpretable models. These methods can be categorized as non-generative and generative. Non-generative methods use information from a ML model to identify the features most responsible for an outcome for a given input. Approaches like [47, 37, 41] interpret ML model decisions by using derived information to fit a locally interpretable model. Others use the gradient of the ML model parameters to perform feature attribution [59, 60, 52, 54, 50, 1, 51], sometimes by employing a reference distribution for the features [51, 11]. This has the advantage of identifying alternative feature values that when substituted for the observed values would result in a different mode outcome. These methods are limited to small contiguous regions of features with high influence on the target model outcome. In so doing, they can struggle to provide plausible changes of the input that are *actionable* by an user in order to correct a certain output or bias of the model. Generative methods such as [7, 5, 4] propose plausible modifications of the input that change the model decision. However the generated perturbations are usually found in pixel space and thus are bound to masking small regions of the image without necessarily having a semantic meaning. Closest to our work are generative counterfactual explanation methods [26, 9, 15, 53] which synthesize perturbed versions of observed data that result in a corresponding change of the model prediction. While these methods provide *valid* and *proximal* explanations for a model outcome, they fail to provide a *diverse* set of *non-trivial* explanations. Mothilal et al. [38] addressed the diversity problem by introducing a diversity constraint between a set of randomly initialized counterfactuals (DICE). However, DICE shares the same problems as [7, 4] since perturbations are directly performed on the observed feature space, and does not take into account *trivial* explanations.

In this work we propose DiVE, a counterfactual explanation method that generates a *diverse* set of *valid*, *proximal*, *sparse*, and *non-trivial* explanations. Appendix A provides a more exhaustive review of the related work.

## 3 PROPOSED METHOD

We propose DiVE, an explainability method that can interpret a ML model by identifying the latent attributes that have the most effect on its output. Summarized in Figure 1, DiVE uses an encoder, a decoder, and a fixed weight ML model. The ML model could be any function for which we have access to its gradients. In this work, we focus on a binary image classifier in order to produce visual explanations. DiVE consists of two main steps. First, the encoder and the decoder are trained in an unsupervised manner to approximate the data distribution on which the ML model was trained. Unlike PE [53], our encoder-decoder model does not need to train on the same dataset that the ML model was trained on. Second, we optimize a set of vectors $\epsilon_i$ to perturb the latent representation $\mathbf{z}$ generated by the trained encoder. The details of the optimization procedure are provided in Algorithm 1 in the Appendix. We use the following 3 main losses for this optimization: a counterfactual loss $\mathcal{L}_{\mathrm{CF}}$ that attempts to fool the ML model, an proximity loss $\mathcal{L}_{\mathrm{prox}}$ that constrains the explanations with respect to the number of changing attributes, and a diversity loss $\mathcal{L}_{\mathrm{div}}$ that enforces the explainer to generate diverse explanations with only one confounding factor for each of them. Finally, we propose several strategies to mask subsets of dimensions in the latent space to prevent the explainer from producing trivial explanations. Next we explain the methodology in more detail.

### 3.1 OBTAINING MEANINGFUL REPRESENTATIONS.

Given a data sample $\mathbf{x} \in \mathcal{X}$, its corresponding target $y \in \{0, 1\}$, and a potentially biased ML model $f(\mathbf{x})$ that approximates $p(y|\mathbf{x})$, our method finds a perturbed version of the same input $\tilde{\mathbf{x}}$ that produces a desired probabilistic outcome $\tilde{y} \in [0, 1]$, so that $f(\tilde{\mathbf{x}}) = \tilde{y}$. In order to produce semantically meaningful counterfactual explanations, perturbations are performed on a latent representation $\mathbf{z} \in \mathcal{Z} \subseteq \mathbb{R}^d$ of the input $\mathbf{x}$. Ideally, each dimension in $\mathcal{Z}$ represents a different semantic concept of the data, *i.e.*, the different dimensions are *disentangled*.

For training the encoder-decoder architecture we use $\beta$-TCVAE [5] since it has been shown to obtain competitive disentanglement performance [34]. It follows the same encoder-decoder structure as the

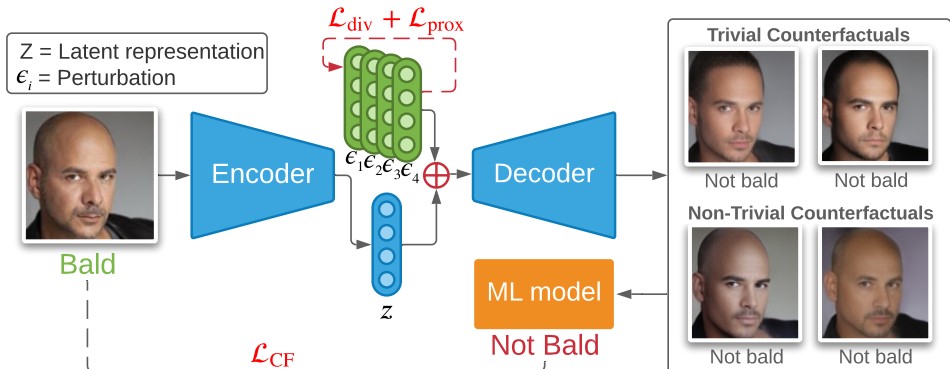

Figure 1: DiVE encodes the input image (left) to explain into a latent representation $z$. Then $z$ is perturbed by $\epsilon$ and decoded as counterfactual examples. During training, $\mathcal{L}_{\text{CF}}$ finds the set of $\epsilon$ that change the ML model classifier outcome while $\mathcal{L}_{\text{div}}$ and $\mathcal{L}_{\text{prox}}$ enforce that the samples are *diverse* while staying *proximal*. These are four valid counterfactuals generated from the experiment in Section 4.4. However, only the bottom row contains counterfactuals where the man is still bald as indicated by the oracle or a human. These counterfactuals identify a weakness in the ML model.

VAE [30], *i.e.*, the input data is first encoded by a neural network $q_\phi(z|\mathbf{x})$ parameterized by $\phi$. Then, the input data is recovered by a decoder neural network $p_\theta(\mathbf{x}|z)$, parameterized by $\theta$. Using a prior $p(z)$ and a uniform distribution over the indexes of the dataset $p(i)$, the original VAE loss is:

$$\mathcal{L}_{VAE} = \mathbb{E}_{p(i)}\mathbb{E}_{q(z|\mathbf{x}_i)}[\log p_\theta(\mathbf{x}_i|z)] - \mathbb{E}_{p(i)}D_{\text{KL}}\left(q_\phi(z|\mathbf{x}_i)||p(z)\right), \tag{1}$$

where the first term is the reconstruction loss and the second is the average divergence from the prior. The core difference of $\beta$-TCVAE is the decomposition of this average divergence:

$$\mathbb{E}_{p(i)}D_{\text{KL}}\left(q_\phi(z|\mathbf{x}_i)||p(z)\right) \rightarrow D_{\text{KL}}\left(q_\phi(z,\mathbf{x}_i)||q_\phi(z)p_\theta(\mathbf{x}_i)\right) + \sum_j D_{\text{KL}}\left(q_\phi(z_j)||p(z_j)\right)$$
$$+ \beta \cdot D_{\text{KL}}\left(q_\phi(z)||\textstyle\prod_j q_\phi(z_j)\right), \tag{2}$$

where the arrow represents a modification of the left terms and equality is obtained when $\beta = 1$. The third term on the right hand side is called total correlation and measures the shared information between all empirical marginals $q_\phi(z_j) = \mathbb{E}_{p(i)}q_\phi(z_j|\mathbf{x}_i)$. By using $\beta > 1$, this part is amplified and encourages further decorrelations between the latent variables and leads to better disentanglement.

In addition to $\beta$-TCVAE, we use the perceptual reconstruction loss from Hou et al. [20]. This replaces the pixel-wise reconstruction loss in Equation 1 by a perceptual reconstruction loss, using the hidden representation of a pre-trained neural network $R$. Specifically, we learn a decoder $D_\theta$ generating an image *i.e.*, $\tilde{\mathbf{x}} = D_\theta(\mathbf{z})$, and this image is re-encoded in a hidden representation: $\boldsymbol{h} = R(\tilde{\mathbf{x}})$, and compared to the original image in the same space using a normal distribution. The reconstruction loss of Equation 1 now becomes:

$$\mathbb{E}_{p(i)}\mathbb{E}_{q(\mathbf{z}|\mathbf{x}_i)}[\log \mathcal{N}(R(\mathbf{x}_i); R(D_\theta(\mathbf{z})), \boldsymbol{I})], \tag{3}$$

Once trained, the weights of the encoder-decoder are fixed for the rest of the steps of our algorithm.

## 3.2 INTERPRETING THE ML MODEL

In order to find weaknesses in the ML model, DiVE searches for a collection of $n$ latent perturbation $\{\epsilon_i\}_{i=1}^n$ such that the decoded output $\tilde{\mathbf{x}}_i = D_\theta(\mathbf{z}+\epsilon_i)$ yields a specific response from the ML model, *i.e.*, $f(\tilde{\mathbf{x}}) = \tilde{y}$ for any chosen $\tilde{y} \in [0, 1]$. We optimize $\epsilon_i$'s by minimizing:

$$\mathcal{L}_{\text{DiVE}}(\mathbf{x}, \tilde{y}, \{\epsilon_i\}_{i=1}^n) = \sum_i \mathcal{L}_{\text{CF}}(\mathbf{x}, \tilde{y}, \boldsymbol{\epsilon_i}) + \lambda \cdot \sum_i \mathcal{L}_{\text{prox}}(\mathbf{x}, \boldsymbol{\epsilon_i}) + \alpha \cdot \mathcal{L}_{\text{div}}(\{\boldsymbol{\epsilon_i}\}_{i=1}^n), \tag{4}$$

where $\lambda$, and $\alpha$ determine the relative importance of the losses. The minimization is performed with gradient descent and the complete algorithm can be found in Algorithm 1 in Appendix D. We now describe the different loss terms.

**Counterfactual loss.** The goal of this loss function is to identify a change of latent attributes that will cause the ML model $f$ to change it's prediction. For example, in face recognition, if the classifier

detects that there is a smile present whenever the hair is brown, then this loss function is likely to change the hair color attribute. This is achieved by sampling from the decoder $\tilde{\mathbf{x}} = D_\theta(\mathbf{z} + \boldsymbol{\epsilon})$, and optimizing the binary cross-entropy between the target $\tilde{y}$ and the prediction $f(\tilde{\mathbf{x}})$:

$$\mathcal{L}_{\text{CF}}(\mathbf{x}, \tilde{y}, \boldsymbol{\epsilon}) = \tilde{y} \cdot \log(f(\tilde{\mathbf{x}})) + (1 - \tilde{y}) \cdot \log(1 - f(\tilde{\mathbf{x}})). \tag{5}$$

**Proximity loss.** The goal of this loss function is to constrain the reconstruction produced by the decoder to be similar in appearance and attributes as the input. It consists of the following two terms,

$$\mathcal{L}_{\text{prox}}(\mathbf{x}, \boldsymbol{\epsilon}) = ||\mathbf{x} - \tilde{\mathbf{x}}||_1 + \gamma \cdot ||\boldsymbol{\epsilon}||_1, \tag{6}$$

where $\gamma$ is a scalar weighting the relative importance of the two terms. The first term ensures that the explanations can be related to the input by constraining the input and the output to be similar. The second term aims to identify a sparse perturbation to the latent space $\mathcal{Z}$ that confounds the ML model. This constrains the explainer to identify the least amount of attributes that affect the classifier's decision in order to produce *sparse* explanations.

**Diversity loss.** This loss prevents the multiple explanations of the model from being identical. For instance, if gender and hair color are spuriously correlated with smile, the model should provide images either with different gender or different hair color. To achieve this, we jointly optimize for a collection of $n$ perturbations $\{\boldsymbol{\epsilon}_i\}_{i=1}^n$ and minimize their pairwise similarity:

$$\mathcal{L}_{\text{div}}(\{\boldsymbol{\epsilon}_i\}_{i=1}^n) = \sqrt{\sum_{i \neq j} \left( \frac{\boldsymbol{\epsilon}_i^T}{||\boldsymbol{\epsilon}_i||_2} \frac{\boldsymbol{\epsilon}_j}{||\boldsymbol{\epsilon}_j||_2} \right)^2}. \tag{7}$$

The method resulting of optimizing Eq. 4 (DiVE) results in *diverse* counterfactuals that are more *valid, proximal,* and *sparse.* However, it may still produce *trivial* explanations, such as exaggerating a smile to explain a smile classifier without considering other valuable biases in the ML model such as hair color. While the diversity loss encourages the orthogonality of the explanations, there might still be several latent variables required to represent all variations of smile.

**Beyond trivial counterfactual explanations.** To find *non-trivial* explanations, we propose to prevent DiVE from perturbing the most influential latent factors of $\mathcal{Z}$ on the ML model. We estimate the influence of each of the latent factors with the average Fisher information matrix:

$$\boldsymbol{F} = \mathbb{E}_{p(i)} \mathbb{E}_{q_\phi(\mathbf{z}|\mathbf{x}_i)} \mathbb{E}_{p(y|\mathbf{z})} \nabla_{\mathbf{z}} \ln p(y|\mathbf{z}) \; \nabla_{\mathbf{z}} \ln p(y|\mathbf{z})^T, \tag{8}$$

where $p(y = 1|\mathbf{z}) = f(D_\theta(\mathbf{z}))$, and $p(y = 0|\mathbf{z}) = 1 - f(D_\theta(\mathbf{z}))$. The diagonal values of $\mathbf{F}$ express the relative influence of each of the latent dimensions on the classifier output. Since the most influential dimensions are likely to be related to the main attribute used by the classifier, we propose to prevent Eq. 4 from perturbing them in order to find more surprising explanations. Thus when producing $n$ explanations, we sort $\mathcal{Z}$ by the magnitude of the diagonal, we partition it into $n$ contiguous chunks that will be optimized for each of the explanations. We call this method DiVE_Fisher.

However, DiVE$_{Fisher}$ does not guarantee that the different partitions of $\mathcal{Z}$ all the factors concerning a *trivial* attribute are grouped together. Thus, we propose to partition $\mathcal{Z}$ into subsets of latent factors that interact with each other when changing the predictions of the ML model. Such interaction can be estimated using $\boldsymbol{F}$ as an affinity measure. We use spectral clustering [55] to obtain a partition of $\mathcal{Z}$. This partition is represented as a collection of mask $\{\boldsymbol{m}_i\}_{i=1}^n$, where $\boldsymbol{m}_i \in \{0, 1\}^d$ represents which dimensions of $\mathcal{Z}$ are part of cluster $i$. Finally, these masks are used in Equation 4 to bound each $\boldsymbol{\epsilon}_i$ to its subspace *i.e.,* $\boldsymbol{\epsilon}_i' = \boldsymbol{\epsilon}_i \circ \boldsymbol{m}_i$, where $\circ$ represents element wise multiplication. Since these masks are orthogonal, this effectively replaces $\mathcal{L}_{\text{div}}$. In Section 4, we highlight the benefits of this clustering approach by comparing to other baselines. We call this method DiVE_FisherSpectral.

## 4 EXPERIMENTAL RESULTS

In this section, we evaluate the described methods on 5 different aspects: (1) the *validity* of the generated explanations as well as the ability to discover biases within the ML model and the data (Section 4.1); (2) their *proximity* in terms of FID, latent space closeness, and face verification accuracy (Section 4.2); (3) the sparsity of the generated counterfactuals (Section 4.3); and (4) the ability to identify diverse *non-trivial* explanations for image misclassifications made by the ML model (Section 4.4); (5) the out-of-distribution performance of DiVE (Section 4.4).

Table 1: **Bias detection experiment.** Ratio of generated counterfactuals classified as "Smiling" and "Non-Smiling" for a classifier biased on gender ($f_{\text{biased}}$) and an unbiased classifier ($f_{\text{unbiased}}$). Bold indicates *Overall* closest to *the Ground truth*.

| ML model | model | | Target label | | | | | |
|---|---|---|---|---|---|---|---|
| | | | Smiling | | | Non-Smiling | | |
| | | PE | xGEM+ | DiVE | PE | xGEM+ | DiVE |
| $f_{\text{biased}}$ | Male | 0.52 | 0.94 | 0.89 | 0.18 | 0.24 | 0.16 |
| | Female | 0.48 | 0.06 | 0.11 | 0.82 | 0.77 | 0.84 |
| | Overall | 0.12 | **0.29** | 0.22 | 0.35 | 0.33 | **0.36** |
| | Ground truth | | 0.75 | | | 0.67 | |
| $f_{\text{unbiased}}$ | Male | 0.48 | 0.41 | 0.42 | 0.47 | 0.38 | 0.44 |
| | Female | 0.52 | 0.59 | 0.58 | 0.53 | 0.62 | 0.57 |
| | Overall | **0.07** | 0.13 | 0.10 | 0.08 | 0.15 | **0.07** |
| | Ground truth | | 0.04 | | | 0.00 | |

Table 2: FID of DiVE compared to xGEM [26], Progressive Exaggeration (PE) [53], xGEM trained with our backbone (xGEM+), and DiVE trained without the perceptual loss (DiVE−−)

| Target Attribute | xGEM | PE | xGEM+ | DiVE−− | DiVE |
|---|---|---|---|---|---|
| | | | **Smiling** | | |
| Present | 111.0 | 46.9 | 67.2 | 54.9 | **30.6** |
| Absent | 112.9 | 56.3 | 77.8 | 62.3 | **33.6** |
| Overall | 106.3 | 35.8 | 66.9 | 55.9 | **29.4** |
| | | | **Young** | | |
| Present | 115.2 | 67.6 | 68.3 | 57.2 | **31.8** |
| Absent | 170.3 | 74.4 | 76.1 | 51.1 | **45.7** |
| Overall | 117.9 | 53.4 | 59.5 | 47.7 | **33.8** |

**Experimental Setup.**     As common procedure [26, 9, 53], we perform experiments on the CelebA database [33]. CelebA is a large-scale dataset containing more than 200K celebrity facial images. Each image is annotated with 40 binary attributes such as "Smiling", "Male", and "Eyeglasses". These attributes allow us to evaluate counterfactual explanations by determining whether they could highlight spurious correlations between multiple attributes such as "lipstick" and "smile". In this setup, explainability methods are trained in the training set and ML models are explained on the validation set. The hyperparameters of the explainer are searched by cross-validation on the training set. We use the same train and validaton splits as PE [53]. Explainers do not have access to the labeled attributes during training.

We test the out-of-distribution (OOD) performance of DiVE with the Synbols dataset [31]. Synbols is an image generator with characters from the Unicode standard and the wide range of artistic fonts provided by the open font community. This provides us to better control on the features present in each set when compared to CelebA. We generate 100K black and white of $32 \times 32$ images from 48 characters in the latin alphabet and more than 1K fonts. We use the character type to create disjoint sets for OOD training and we use the fonts to introduce biases in the data. We provide a sample of the dataset in Figure 8 in Appendix I.

We compare four versions of our method to three existing methods. DiVE, resulting of optimizing Eq. 4. DiVE$_{\text{Fisher}}$, which extends DiVE by using the Fisher information matrix introduced in Eq. 8. DiVE$_{\text{FisherSpectral}}$, which extends DiVE$_{\text{Fisher}}$ with spectral clustering. We introduce two additional ablations of our method, DiVE−− and DiVE$_{\text{Random}}$. DiVE−− is equivalent to DiVE but using a pixel-based reconstruction loss instead of the perceptual loss. DiVE$_{\text{Random}}$ uses random masks instead of using the Fisher information. Finally, we compare our baselines with xGEM as described in Joshi et al. [26], xGEM+, which is the same as xGem but uses the same auto-encoding architecture as DiVE, and PE as described by Singla et al. [53]. For our methods, we provide implementation details, architecture description, and algorithm in Appendix D.

## 4.1 VALIDITY AND BIAS DETECTION

We evaluate DiVE's ability to detect biases in the data. We follow the same procedure as PE [53], and train two binary classifiers for the attribute "Smiling". The first one is trained on a biased version of CelebA where all the male celebrities are smiling and all the female are not smiling ($f_{biased}$). The second one is trained on the unbiased version of the data ($f_{unbiased}$). Both classifiers are evaluated on the CelebA validation set. Also following Singla et al. [53], we train an oracle classifier ($f_{\text{oracle}}$) based on VGGFace2 [3] which obtains perfect accuracy on the gender attribute. The hypothesis is that if "Smiling" and gender are confounded by the classifier, so should be the explanations. Therefore, we could identify biases when the generated examples not only change the target attribute but also the confounded one. To generate the counterfactuals, DiVE produces perturbations until it changes the original prediction of the classifier (*e.g.* "Smiling" to "Non-Smiling").

We follow the procedure introduced in [26, 53] and report a confounding metric for bias detection in In Table 1. The columns *Smiling* and *Non-Smiling* indicate the target class for counterfactual generation. The rows *Male* and *Female* contain the proportion of counterfactuals that are classified by the oracle as *Male* and *Female*. We can see that the generated explanations for $f_{\text{biased}}$ are classified

Table 3: Average number attributes changed per explanation and percentage of non-trivial explanations. This experiment evaluates the counterfactuals generated by different methods for a ML model trained on the attribute 'Young' of the CelebA dataset. xGEM++ is xGEM+ using $\beta$-TCVAE as generator.

| | PE [53] | xGEM+ [26] | xGEM++ | DiVE | DiVE$_{Fisher}$ | DiVE$_{FisherSpectral}$ |
|---|---|---|---|---|---|---|
| Attr. change | 03.74 | 06.92 | 06.70 | 04.81 | 04.82 | 04.58 |
| Non-trivial (%) | 05.12 | 18.56 | 34.62 | 43.51 | 42.99 | 51.07 |

more often as *Male* when the target attribute is *Smiling* and *Female* when the target attribute is *Non-Smiling*. The confounding metric, denoted as overall, is the fraction of generated explanations for which the gender was changed with respect to the original image. It thus reflect the magnitude of the the bias as approximated by the explainers.

Singla et al. [53] consider that a model is better than another if the confounding metric is the highest on $f_{biased}$ and the lowest on $f_{unbiased}$. However, they assume that $f_{biased}$ always predicts the *Gender* based on *Smile*. Instead, we propose to evaluate the confounding metric by comparing it to the empirical bias of the model, denoted as ground truth in the Table 1. Details provided in Appendix J.

We observe that DiVE is more successful than PE at detecting biases although the generative model of DiVE was not trained with the biased data. While xGEM+ has a higher success rate at detecting biases in some cases, it produces lower-quality images that are far from the input. In Figure 5 in Appendix B, we provide samples generated by our method with the two classifiers and compare them to PE and xGEM+. We found that gender changes with the "Smiling" attribute with $f_{biased}$ while for $f_{unbiased}$ it stayed the same. In addition, we also observed that for $f_{biased}$ the correlation between "Smile" and "Gender" is higher than for PE. It can also be observed that xGEM+ fails to retain the identity of the person in x when compared to PE and our method.

## 4.2 COUNTERFACTUAL EXPLANATION PROXIMITY

We evaluate the *proximity* of the counterfactual explanations using FID scores [19] as described by Singla et al. [53]. The scores are based on the target attributes "Smiling" and "Young", and are divided into 3 categories: *Present*, *Absent*, and *Overall*. *Present* considers explanations for which the ML model outputs a probability greater than 0.9 for the target attribute. *Absent* refers to explanations with a probability lower than 0.1. *Overall* considers all the successful counterfactuals, which changed the original prediction of the ML model.

We report these scores in Table 2 for all 3 categories. DiVE produces the best quality counterfactuals, surpassing PE by 6.3 FID points for the "Smiling" target and 19.6 FID points for the "Young" target in the *Overall* category. DiVE obtains lower FID than xGEM+ which shows that the improvement not only comes from the superior architecture of our method. Further, there are two other factors that explain the improvement of DiVE's FID. First, the $\beta$-TCVAE decomposition of the KL divergence improves the disentanglement ability of the model while suffering less reconstruction degradation than the VAE. Second, the perceptual loss makes the image quality constructed by DiVE to be comparable with that of the GAN used in PE. In addition, Table 4 in the Appendix shows that DiVE is more successful at preserving the identity of the faces than PE and xGEM and thus at producing feasible explanations. These results suggest that the combination of disentangled latent features and the regularization of the latent features help DiVE to produce the minimal perturbations of the input that produce a successful counterfactual.

In Figure 5 in Appendix B we show qualitative results obtained by targeting different probability ranges for the output of the ML model as described in PE. As seen in Figure 5, DiVE produces more natural-looking facial expressions than xGEM+ and PE. Additional results for "Smiling" and "Young" are provided in Figures 3 and 4 in the Appendix B.

## 4.3 COUNTERFACTUAL EXPLANATION SPARSITY

Explanations that produce sparse changes in the attributes of the image are more probable to be actionable. In this section we quantitatively compare the amount of valid and sparse counterfactuals provided by different baselines. Table 3 shows the results for a classifier model trained on the at-

tribute Young of the CelebA dataset.[1] The first row shows the number of attributes that each method change in average to generate a valid counterfactual. Methods that require to change less attributes are likely to be more actionable. We observe that DiVE changes less attributes on average than xGEM+. We also observe that DiVE$_{\text{FisherSpectral}}$ is the method that changes less attributes among all the baselines. To better understand the effect of disentangled representations, we also report results for a version of xGEM+ with the $\beta$-TCVAE backbone (xGEM++). We do not observe significant effects on the sparsity of the counterfactuals. In fact, a fine-grained decomposition of concepts in the latent space could lead to lower the sparsity.

## 4.4 BEYOND TRIVIAL EXPLANATIONS

Previous works on counterfactual generations tend to produce *trivial* input perturbations to change the output of the ML model. That is, they tend to increase/decrease the presence of the attribute that the classifier is predicting. For instance, in Figure 5 all the explainers put a smile on the input face in order to increase the probability for "smile". While that is correct, this explanation does not provide much insight about the potential weaknesses of the ML model. Instead, in this work we emphasize producing non-trivial explanations, that are different from the main attribute that the ML model has been trained to identify. These kind of explanations provide more insight about the factors that affect the classifier and thus provide cues on how to improve the model or how to fix incorrect predictions.

To evaluate this, we propose a new benchmark that measures a method's ability to generate valuable explanations. For an explanation to be valuable, it should 1) be misclassified by the ML model (*valid*), 2) not modify the main attribute being classified (*non-trivial*), and 3) not have diverged too much from the original sample (*proximal*). A misclassification provides insights into the weaknesses of the model. However, the counterfactual is even more insightful when it stays close to the original image as it singles-out spurious correlations learned by the ML model. Because it is costly to provide human evaluation of an automatic benchmark, we approximate both the proximity and the real class with the VGGFace2-based oracle. We choose the VGGFace2 model as it is less likely to share the same biases as the ML model, since it was trained for a different task than the ML model with an order of magnitude more data. We conduct a human evaluation experiment in Appendix F, and we find a significant correlation between the oracle and the human predictions. For 1) and 2) we deem that an explanation is successful if the ML model and the oracle make different predictions about the counterfactual. *E.g.*, the top counterfactuals in Figure 1 are not deemed successful explanations because both the ML model and the oracle agree on its class, however the two in the bottom row are successful because only the oracle made the correct prediction. These explanations where generated by DiVE$_{\text{FisherSpectral}}$. As for 3) we measure the proximity with the cosine distance between the sample and the counterfactual in the feature space of the oracle.

We test all methods from Section 4 on a subset of the CelebA validation set described in Appendix E. We report the results of the full hyperparameter search (see Appendix E) in Figure 2a. The vertical axis shows the success rate of the explainers, *i.e.*, the ratio of valid explanations that are non-trivial. This is the misclassification rate of the ML model on the explanations. The dots denote the mean performances and the curves are computed with Kernel Density Estimation (KDE). On average, DiVE improves the similarity metric over xGEM+ highlighting the importance of disentangled representations for identity preservation. Moreover, using information from the diagonal of the Fisher Information Matrix as described in Eq. 8 further improves the explanations as shown by the higher success rate of DiVE$_{\text{Fisher}}$ over DiVE and DiVE$_{\text{Random}}$. Thus, preventing the model from perturbing the most influential latent factors helps to uncover spurious correlations that affect the ML model. Finally, the proposed spectral clustering of the full Fisher Matrix attains the best performance validating that the latent space partition can guide the gradient-based search towards better explanations. We reach the same conclusions in Table 3, where we provide a comparison with PE for the attribute *Young*. In addition, we provide results for a version of xGEM+ with more disentangled latent factos (xGEM++). We find that disentangled representations provide the explainer with a more precise control on the semantic concepts being perturbed, which increases the success rate of the explainer by 16%.

**Out-of-distribution generalization.** In the previous experiments, the generative model of DiVE was trained on the same data distribution (*i.e.*, CelebA faces) as the ML model. We test the out-

---

[1] The code and pre-trained models of PE are only available for the attribute Young.

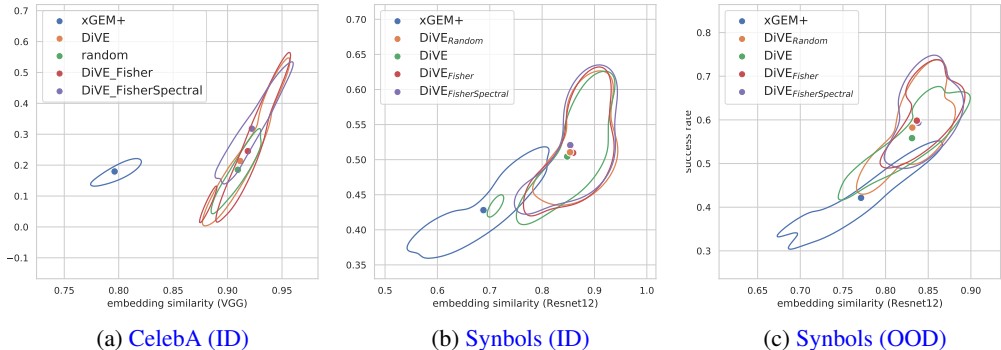



(a) CelebA (ID)  (b) Synbols (ID)  (c) Synbols (OOD)



Figure 2: **Beyond trivial explanations.** The rate of successful explanations (y-axis) plotted against embedding similarity (x-axis) for all methods. For both metrics, higher is better, *i.e.*, the most valuable explanations are in the top-right corner. For each method, we ran an hyperparameter sweep and denote the mean of the performances with a dot. The curves are computed with KDE. The left plot shows the performance on CelebA and the other two plots shows the performance for in-distribution (ID) and out-of-distribution (OOD) experiments on Synbols . All DiVE methods outperform xGEM+ on both metrics simultaneously when conditioning on *successful counterfactuals*. In both experiments, DiVE_Fisher and DiVE_FisherSpectral improve the performance over both DiVE_Random and DiVE.

.

of-distribution performance of DiVE by training its auto-encoder on a subset of the latin alphabet of the Synbols dataset [31]. Then, counterfactual explanations are produced for a different disjoint subset of the alphabet. To evaluate the effectiveness of DiVE in finding biases on the ML model, we introduce spurious correlations in the data. Concretely, we assign different fonts to each of the letters in the alphabet as detailed in Appendix I. In-distribution (ID) results are reported in Figure 2b for reference, and OD results are reported in Figure 2c. We observe that DiVE is able to find valuable countefactuals even when the VAE was not trained on the same data distribution. Moreover, results are consistent with the CelebA experiment, with DiVE outperforming xGEM+ and Fiser information-based methods outperforming the rest.

## 5 LIMITATIONS AND FUTURE WORK

This work shows that a good generative model can provide interesting insights on the biases of a ML model. However, this relies on a properly disentangled representation. In the case where the generative model would be heavily entangled it would fail to produce explanations with a sparse amount of features. However, our approach can still tolerate a small amount of entanglement, yielding a small decrease in interpretability. We expect that progress in identifiability [35, 28] will increase the quality of representations. With a perfectly disentangled model, our approach could still miss some explanations or biases. *E.g.*, with the spectral clustering of the Fisher, we group latent variables and only produce a single explanation per group in order to present explanations that are conceptually different. This may leave behind some important explanations, but the user can simply increase the number of clusters or the number of explanation per clusters for a more in-depth analysis.

In addition to the challenge of achieving disentangled representations, finding the optimal hyperparameters for the VAE and their generalization out of the training distribution is an open problem. Moreover, if the generative model is trained on biased data, one could expect the counterfactuals to be biased as well. However, as we show in Figure 2c, our model still finds non-trivial explanations when applied out of distribution. In that way, it could be trained on a larger unlabeled dataset to overcome possible biases caused by the lack of annotated data.

Although the generative model plays an important role to produce actionable counterfactuals in the computer vision domain domain, our work could be extended to other domains. For example, Eq. 4 could be applied to find non-trivial explanations on tabular data by directly optimizing the observed features instead of the latent factors of the VAE. However, further work would be needed to adapt the DiVE loss functions to produce perturbations on discrete and categorical variables.

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

## APPENDIX

## A  EXTENDED RELATED WORK

Counterfactual explanation lies inside a more broadly-connected body of work for explaining classifier decisions. Different lines of work share this goal but vary in the assumptions they make about what elements of the model and data to emphasize as way of explanation.

**Model-agnostic counterfactual explanation.**   Like [47, 37], these models make no assumptions about model structure, and interact solely with its label predictions. Karimi et al. [27] develop a model agnostic, as well as metric agnostic approach. They reduce the search for counterfactual explanations (along with user-provided constraints) into a series of satisfiability problems to be solved with off-the-shelf SAT solvers. Similar in spirit to [47], Guidotti et al. [16] first construct a local neighbourhood around test instances, finding both positive and negative exemplars within the neighbourhood. These are used to learn a shallow decision tree, and explanations are provided in terms of the inspection of its nodes and structure. Subsequent work builds on this local neighbourhood idea [17], but specializes to medical diagnostic images. They use a VAE to generate both positive and negative samples, then use random heuristic search to arrive at a balanced set. The generated explanatory samples are used to produce a saliency feature map for the test data point by considering the median absolute deviation of pixel-wise differences between the test point, and the positive and negative example sets.

**Gradient based feature attribution.**   These methods identify input features responsible for the greatest change in the loss function, as measured by the magnitude of the gradient with respect to the inputs. Early work in this area focused on how methodological improvements for object detection in images could be re-purposed for feature attribution [59, 60], followed by work summarized gradient information in different ways [52, 54, 50]. Closer inspection identified pitfalls of gradient-based methods, including induced bias due to gradient saturation or network structure [1], as well as discontinuity due to activation functions [51]. These methods typically produce dense feature maps, which are difficult to interpret. In our work we address this by constraining the generative process of our counterfactual explanations.

**Reference based feature attribution.**   These methods focus instead on measuring the differences observed by substituting observed input values with ones drawn from some reference distribution, and accumulating the effects of these changes as they are back-propagated to the input features. Shrikumar et al. [51] use a modified back-propagation approach to gracefully handle zero gradients and negative contributions, but leave the reference to be specified by the user. Fong & Vedaldi [11] propose three different heuristics for reference values: replacement with a constant, addition of noise, and blurring. Other recent efforts have focused on more complex proposals of the reference distribution. Chen et al. [5] construct a probabilistic model that acts as a lower bound on the mutual information between inputs and the predicted class, and choose zero values for regions deemed uninformative. Building on desiderata proposed by Dabkowski & Gal [7], Chang et al. [4] use a generative model to marginalize over latent values of relevant regions, drawing plausible values for each. These methods typically either do not identify changes that would *alter* a classifier decision, or they do not consider the plausibility of those changes.

**Counterfactual explanations.**   Rather than identify a set of features, counterfactual explanation methods instead generate perturbed versions of observed data that result in a corresponding change in model prediction. These methods usually assume both more access to  model output and parameters, as well as constructing a generative model of the data to find trajectories of variation that elucidate model behaviour for a given test instance.

Joshi et al. [26] propose a gradient guided search in latent space (via a learned encoder model), where they progressively take gradient steps with respect to a regularized loss that combines a term for plausibility of the generated data, and the loss of the ML model. Denton et al. [9] use a Generative Adversarial Network (GAN) [14] for detecting bias present in multi-label datasets. They modify the generator to obtain latent codes for different data points and learn a linear decision boundary in the latent space for each class attribute. By sampling generated data points along the vector orthogonal

to the decision boundary, they observe how crossing the boundary for one attribute causes undesired changes in others. Some counterfactual estimation methods forego a generative model by instead solving a surrogate editing problem. Given an original image (with some predicted class), and an image with a desired class prediction value, Goyal et al. [15] produce a counterfactual explanation through a series of edits to the original image by value substitutions in the learned representations of both images. Similar in spirit are Dhurandhar et al. [10] and Van Looveren & Klaise [57]. The former propose a search over features to highlight subsets of those present in each test data point that are typically present in the assigned class, as well as features usually absent in examples from adjacent classes (instances of which are easily confused with the label for the test point predicted by the model). The latter generate counterfactual data that is proximal to $x_test$, with a sparse set of changes, and close to the training distribution. Their innovation is to use class prototypes to serve as an additional regularization term in the optimization problem whose solution produces a counterfactual.

Several methods go beyond providing counterfactually generated data for explaining model decisions, by additionally qualifying the effect of proposed changed between a test data point and each counterfactual. Mothilal et al. [38] focus on tabular data, and generate sets of counterfactual explanations through iterative gradient based improvement, measuring the cost of each counterfactual by either distance in feature space, or the sparsity of the set of changes (while also allowing domain expertise to be applied). Poyiadzi et al. [44] construct a weighted graph between each pair of data point, and identify counterfactuals (within the training data) by finding the shortest paths from a test data point to data points with opposing classes. Pawelczyk et al. [42] focus on modelling the density of the data to provide 'attainable' counterfactuals, defined to be proximal to test data points, yet not lying in low-density sub-spaces of the data. They further propose to weigh each counterfactual by the changes in percentiles of the cumulative distribution function for each feature, relative to the value of a test data point.

# B  QUALITATIVE RESULTS

In Figure 5 we show qualitative results obtained by targeting different probability ranges for the output of the ML model as described in PE. Note that PE directly optimizes the generative model to take an input variable $\delta \in \mathbb{R}$ that defines the desired output probability $\tilde{y} = f(\mathbf{x}) + \delta$. To obtain explanations at different probability targets, we train a second order spline on the trajectory of perturbations produced during the gradient descent steps of our method. Thus, given the set of perturbations $\{\epsilon_t\}$, $\forall t \in 1..\tau$, obtained during $\tau$ gradient steps, and the corresponding black-box outputs $\{f(y|\epsilon_t)\}$, the spline obtains the $\epsilon_{\tilde{y}}$ for a target output $\tilde{y}$ by interpolation. As seen in Figure 5, DiVE produces more natural-looking facial expressions than xGEM+ and PE. Although DiVE is not explicitly trained to produce exemplars at intermediate target probabilities, our explanations are more correlated with the target probabilities than PE. Additional results for "Smiling" and "Young" are provided in Figure 3,4.

Figure 3,4 present counterfactual explanations for additional persons and attributes. The results show that DiVE achieves higher quality reconstructions compared to other methods. Further, the reconstructions made by DiVE are more correlated with the desired target for the ML model output $f(x)$. In Figure 5, we provide samples generated by our method with a gender-biased classifier $f_{\text{biased}}$ and an unbiased classifier $f_{\text{unbiased}}$. We compare DiVE to PE and xGEM+. We found that gender changes with the "Smiling" attribute with $f_{\text{biased}}$ while for $f_{\text{unbiased}}$ it stayed the same. In addition, we also observed that for $f_{\text{biased}}$ the correlation between "Smile" and "Gender" is higher than for PE. It can also be observed that xGEM+ fails to retain the identity of the person in $\mathbf{x}$ when compared to PE and our method. Finally, Figure 6 shows *successful counterfactuals* for different instantiations of DiVE.

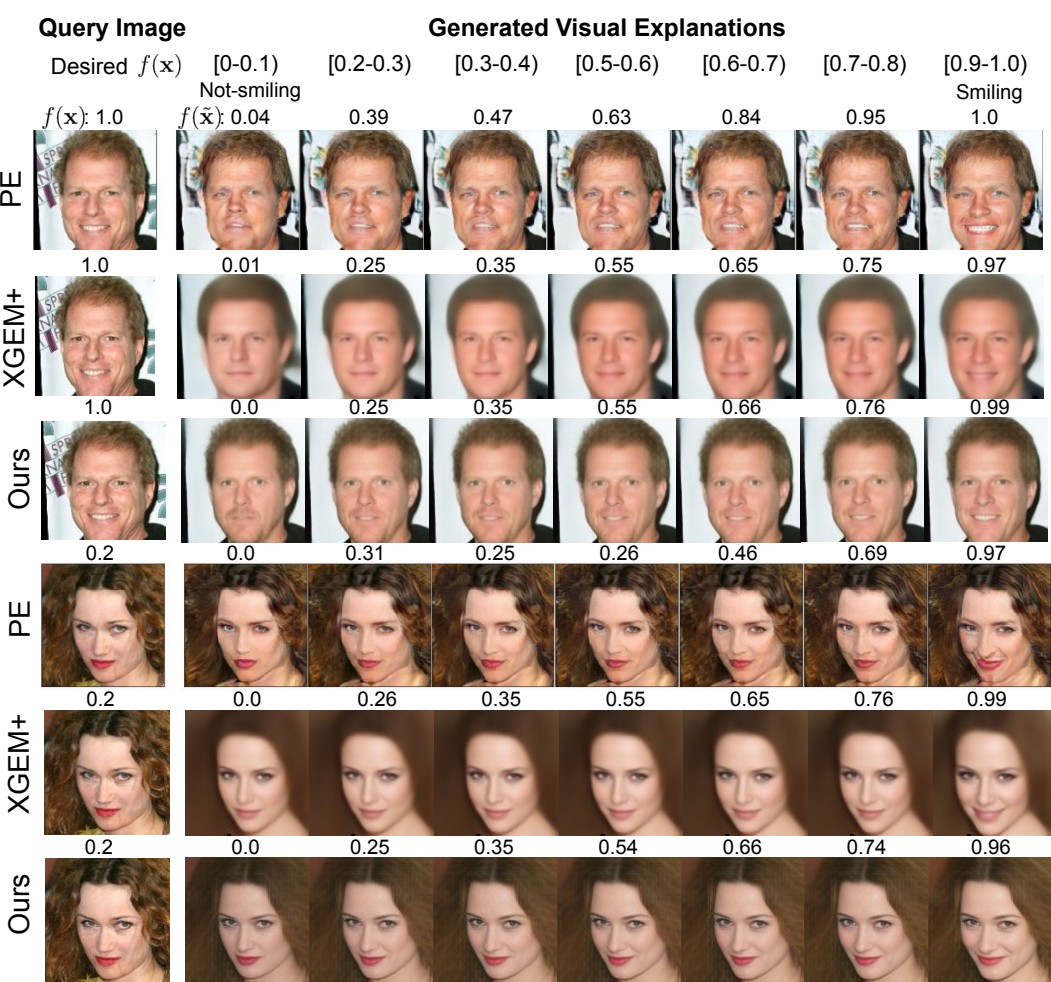

Figure 3: Qualitative results of DiVE, Progressive Exaggeration (PE) [53], and xGEM [26] for the "Smiling" attribute. Each column shows the explanations generated for a target probability output of the ML model. The numbers on top of each row show the actual output of the ML model.

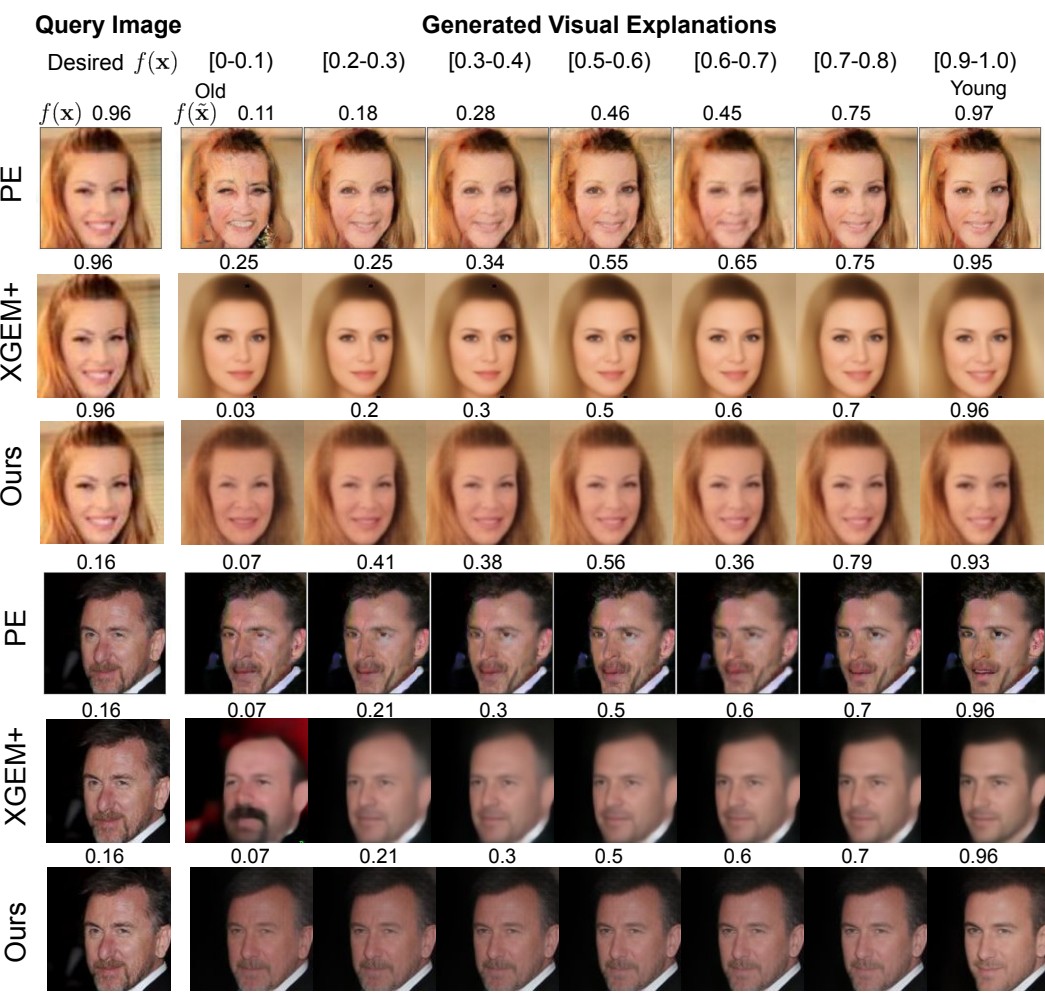

Figure 4: **Qualitative results** of DiVE, Progressive Exaggeration (PE) [53], and xGEM+ for the "Young" attribute. Each column shows the explanations generated for a target probability output of the ML model. The numbers on top of each row show the actual output of the ML model.

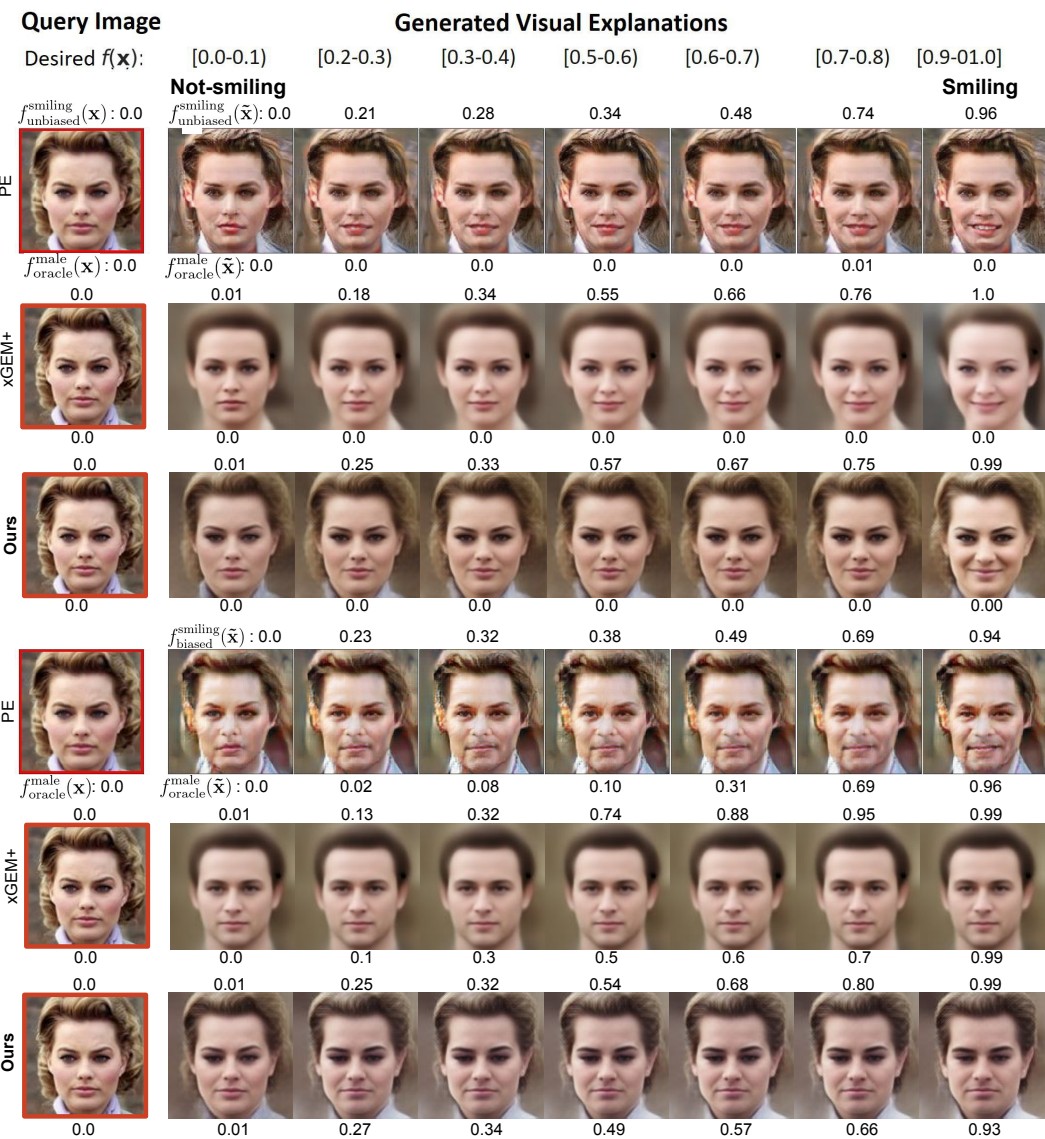

Figure 5: **Bias detection experiment.** Each column presents an explanation for a target "Smiling" probability interval. Rows contain explanations produced by PE [53], xGEM+ and our DiVE. (a) of a gender-unbiased classifier, and (b) corresponds to explanations of a gender-biased "Smile" classifier. The classifier output probability is displayed on top of the images while the oracle prediction for gender is displayed at the bottom.

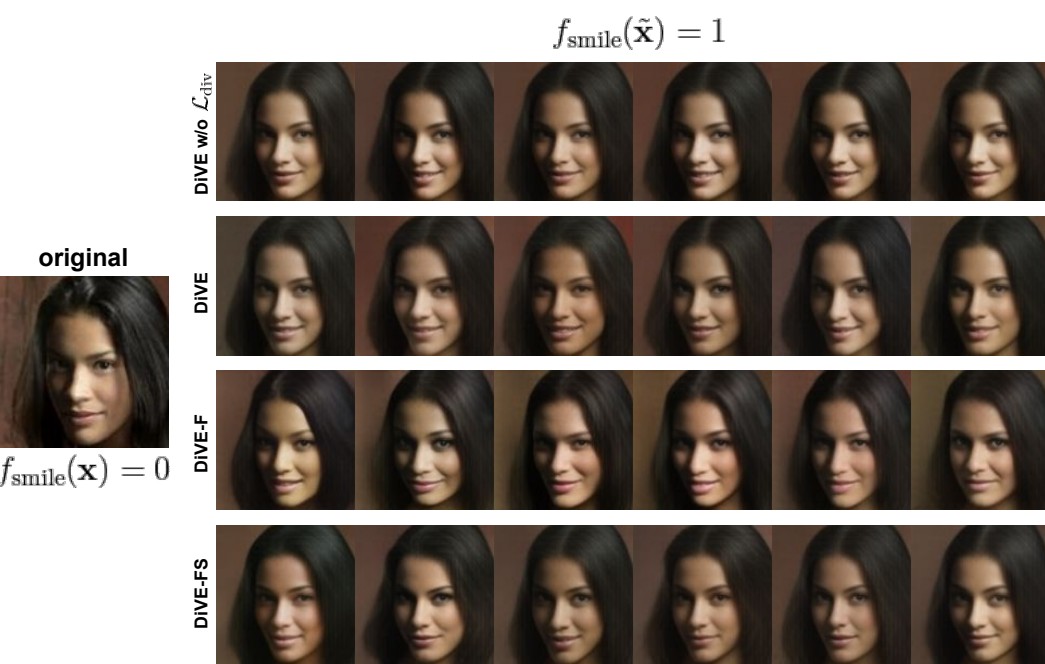

Figure 6: Successful counterfactual generations for different instantions of DiVE. Here, the original image was misclassified as non-smiling. All methodologies were able to correctly add a smile to the woman.

## C  IDENTITY PRESERVATION

As argued, valuable explanations should remain proximal to the original image. Accordingly, performed the identity preservation experiment found in [53] to benchmark the methodologies against each other. Specifically, use the VGGFace2-based [3] oracle to extract latent codes for the original images as well as for the explanations and report **latent space closeness** as the fraction of time the explanations' latent codes are the closest to their respective original image latent codes' compared to the explanations on different original images. Further, we report **face verification** accuracy which consist of the fraction of time the cosine distance between the aforementioned latent codes is below 0.5.

Table 4 presents both metrics for DiVE and its baselines on the "Smilling" and "Young" classification tasks. We find that DiVE outperforms all other methods on the "Young" classification task and almost all on the "Smiling" task.

| | CelebA:Smiling | | | | CelebA:Young | | | |
|---|---|---|---|---|---|---|---|---|
| | xGEM | PE | xGEM+ | DiVE (ours) | xGEM | PE | xGEM+ | DiVE (ours) |
| **Latent Space Closeness** | 88.2 | 88.0 | **99.8** | 98.7 | 89.5 | 81.6 | 97.5 | **99.1** |
| **Face Verification Accuracy** | 0.0 | 85.3 | 91.2 | **97.3** | 0.0 | 72.2 | 97.4 | **98.2** |

Table 4: Identity preserving performance on two prediction tasks.

# D   IMPLEMENTATION DETAILS

In this Section, we provide provide the details to ensure the that our method is reproducible.

**Architecture details.**   DiVE's architecture is a variation BigGAN [2] as shown in Table 6. We chose this architecture because it achieved impressive FID results on the ImageNet [8]. The decoder (Table 6b) is a simplified version of the $128 \times 128$ BigGAN's residual generator, without non-local blocks nor feature concatenation. We use InstanceNorm [56] instead of BatchNorm [23] to obtain consistent outputs at inference time without the need of an additional mechanism such as recomputing statistics [2]. All the InstanceNorm operations of the decoder are conditioned on the input code **z** in the same way as FILM layers [43]. The encoder (Table 6a) follows the same structure as the BigGAN $128 \times 128$ discriminator with the same simplifications done to our generator. We use the Swish non-linearity [46] in all layers except for the output of the decoder, which uses a Tanh activation.

For all experiments we use a latent feature space of 128 dimensions. The ELBO has a natural principled way of selecting the dimensionality of the latent representation. If d is larger than necessary, it will not enhance the reconstruction error and the optimization of the ELBO will make the posterior equal to the prior for these extra dimensions. More can be found on the topic in [36]. In practice, we experimented with $d = \{64, 128, 256\}$ and found that with $d = 128$ we achieved a slightly lower ELBO.

To project the 2d features produced by the encoder to a flat vector $(\mu, \log(\sigma^2))$, and to project the sampled codes **z** to a 2d space for the decoder, we use 3-layer MLPs. For the face attribute classifiers, we use the same DenseNet [21] architecture as described in Progressive Exaggeration [53].

**Optimization details.**   All the models are optimized with Adam [29] with a batch size of 256. During the training step, the auto-encoders are optimized for 400 epochs with a learning rate of $4 \cdot 10^{-4}$. The classifiers are optimized for 100 epochs with a learning rate of $10^{-4}$. To prevent the auto-encoders from suffering KL vanishing, we adopt the cyclical annealing schedule proposed by Fu et al. [12] on the third term of Equation 2.

**Counterfactual inference details.**   At inference time, the perturbations are optimized with Adam until the ML model output for the generated explanation $f(\tilde{\mathbf{x}})$ only differs from the target output $\tilde{y}$ by a margin $\delta$ or when a maximum number of iterations $\tau$ is reached. We set $\tau = 20$ for all the experiments since more than 90% of the counterfactuals are found after that many iterations. The different $\boldsymbol{\epsilon}_i$ are initialized by sampling from a normal distribution $\mathcal{N} \sim (0, 0.01)$. For the DiVE$_{Fisher}$ baseline, to identify the most valuable explanations, we sort $\boldsymbol{\epsilon}$ by the magnitude of $\mathbf{f} = \text{diag}(\boldsymbol{F})$. Then, we divide the dimensions of the sorted $\boldsymbol{\epsilon}$ into $N$ contiguous partitions of size $k = \frac{D}{N}$, where $D$ is the dimensionality of $\mathcal{Z}$. Formally, let $\boldsymbol{\epsilon}^{(\mathbf{f})}$ be $\boldsymbol{\epsilon}$ sorted by $\mathbf{f}$, then $\boldsymbol{\epsilon}^{(\mathbf{f})}$ is constrained as follows,

$$\epsilon_{i,j}^{(\mathbf{f})} = \begin{cases} 0, & \text{if } j \in [(i-1) \cdot k, i \cdot k] \\ \epsilon_{i,j}^{(\mathbf{f})}, & \text{otherwise} \end{cases}, \tag{9}$$

where $i \in 1..N$ indexes each of the multiple $\boldsymbol{\epsilon}$, and $j \in 1..D$ indexes the dimensions of $\boldsymbol{\epsilon}$. As a result we obtain partitions with different order of complexity. Masking the first partition results in explanations that are most implicit within the model and the data. On the other hand, masking the last partition results in explanations that are more explicit.

To compare with Singla et al. [53] in Figure 3-5 we produced counterfactuals at arbitrary target values $\tilde{y}$ of the output of the ML model classifier. One way to achieve this would be to optimize $\mathcal{L}_{\text{CF}}$ for each of the target probabilities. However, these successive optimizations would slow down the process of counterfactual generation. Instead, we propose to directly maximize the target class probability and then interpolate between the points obtained in the gradient descent trajectory to obtain the latent factors of the different target probabilities. Thus, given the set of perturbations $\{\boldsymbol{\epsilon}_t\}$, $\forall t \in 1..\tau$, obtained during $\tau$ gradient steps, and the corresponding ML model outputs $\{f(y|\boldsymbol{\epsilon}_t)\}$, we obtain the $\boldsymbol{\epsilon}_{\tilde{y}}$ for a target output $\tilde{y}$ by interpolation. We do such interpolation by fitting a piecewise quadratic polynomial on the latent trajectory, commonly known as Spline in the computer graphics literature.

# E  BEYOND TRIVIAL EXPLANATIONS EXPERIMENTAL SETUP

The experimental benchmark proposed in Section 4.4 is performed on a subset of the validation set of CelebA. This subset is composed of 4 images for each CelebA attribute. From these 4 images, 2 were correctly classified by the ML model, while the other 2 were misclassified. The two correctly classified images are chosen so that one was classified with a high confidence of 0.9 and the other one with low confidence of 0.1. The 2 misclassifications were chosen with the same criterion. The total size of the dataset is of 320 images. For each of these images we generate $k$ counterfactual explanations. From these counterfactuals, we report the ratio of successful explanations.

Here are the specific values we tried in our hyperparameter search: $\gamma \in [0.0, 0.001, 0.1, 1.0]$, $\alpha \in [0.0, 0.001, 0.1, 1.0]$, $\lambda \in [0.0001, 0.0005, 0.001]$, number of explanations 2 to 15 and learning rate $\in [0.05, 0.1]$. Be xGEM+ doesn't have a $\gamma$ nor $\alpha$ parameter, we increased its learning rate span to $[0.01, 0.05, 0.1]$ to reduce the gap in its search space compared with DiVE. We also changed the ranomd seeds and ran a total of 256 trials.

## F  Human Evaluation

| Method | Human $\neq$ ML Classifier (real non-trivial) | Correlation | p-value |
|---|---|---|---|
| xGEM+ [26] | 38.37% | 0.37 | 0.000 |
| DiVE | 38.65% | 0.25 | 0.002 |
| DiVE$_{Random}$ | 38.89% | 0.24 | 0.001 |
| DiVE$_{Fisher}$ | 40.56% | 0.17 | 0.023 |
| DiVE$_{FisherSpectral}$ | **41.90**% | 0.23 | 0.001 |

Table 5: Human evaluation. The first column contains the percentage of non-trivial counterfactuals from the perspective of the human oracle. These counterfactuals confuse the ML classifier without changing the main attribute being classified from the perspective of a human. The second column contains the Pearson correlation between the human and the oracle's predictions. The third column contains the p-value for a t-test with the null hypothesis of the human and oracle predictions being uncorrelated.

We built a web-based human evaluation task to assess if DiVE is more successful at finding *non-trivial* counterfactuals than previous state of the art and the effectiveness of the VGG-based oracle, see Figure 7. For that, we present humans with valid counterfactuals and ask them whether the main attribute being classified by the ML model is present in the image or not. We use a subset of CelebA containing a random sample of 4 images per attribute, each one classified by the VGG$_{Face}$ oracle as containing the attribute with the following levels of confidence: $[0.1, 0.4, 0.6, 0.9]$. From each of these 160 images, we generated counterfactuals with xGEM+ [26], DiVE, DiVE$_{Random}$, DiVE$_{Fisher}$, and DiVE$_{FisherSpectral}$ and show the valid counterfactuals to the human annotators. Results are reported in Table 5. In the left column we observe that leveraging the Fisher information results in finding more non-trivial counterfactuals, which confuse the ML model without changing the main attribute being classified. In the second column we report the Pearson correlation between the oracle and the classifier predictions. A statistical inference test reveals a significant correlation (p-value$\leq$0.02).

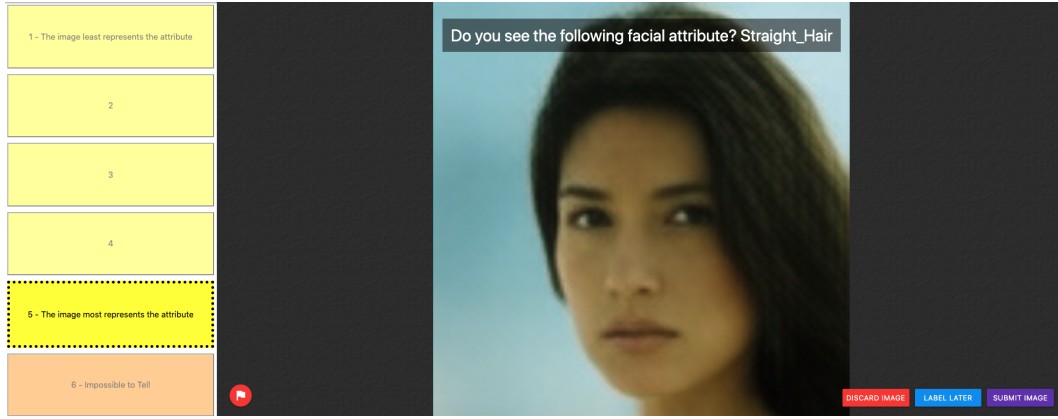

Figure 7: Labelling interface. The user is presented with a counterfactual image and has to choose if the target attribute is present or not in the image.

## G  MODEL ARCHITECTURE

Table 6 presents the architecture of the encoder and decoder used in DiVE.

Table 6: DiCe architecture for $128 \times 128$ images. $ch$ represents the channel width multiplier in each network.

| RGB image $x \in \mathbb{R}^{128 \times 128 \times 3}$ |
|:---:|
| ResBlock down $3ch \to 16ch$ |
| ResBlock $16ch \to 32ch$ |
| ResBlock down $32ch \to 32ch$ |
| ResBlock $32ch \to 64ch$ |
| ResBlock down $64ch \to 64ch$ |
| ResBlock $64ch \to 128ch$ |
| ResBlock down $128ch \to 128ch$ |
| ResBlock $128ch \to 128ch$ |
| ResBlock down $128ch \to 128ch$ |
| IN, Swish, Linear $128ch \times 4 \times 4 \to 128ch$ |
| IN, Swish, Linear $128ch \to 128ch$ |
| IN, Swish, Linear $128ch \to 128ch \times 2$ |
| $z \sim \mathcal{N}(\mu \in \mathbb{R}^{128}, \sigma \in \mathbb{R}^{128})$ |

(a) Encoder

| $z \in \mathbb{R}^{128}$ |
|:---:|
| Linear $128ch \to 128ch$ |
| Linear $128ch \to 128ch$ |
| Linear $128ch \to 128ch \times 4 \times 4$ |
| ResBlock up $128ch \to 64ch$ |
| ResBlock up $64ch \to 32ch$ |
| ResBlock $32ch \to 16ch$ |
| ResBlock up $16ch \to 16ch$ |
| ResBlock $16ch \to 16ch$ |
| ResBlock up $16ch \to 16ch$ |
| ResBlock $16ch \to 16ch$ |
| IN, Swish, Conv $16ch \to 3$ 
 tanh |

(b) Decoder

# H    Model Algorithm

Algorithm 1 presents the steps needed for DiVE to generate explanations for a given ML model using a sample input image.

---

**Algorithm 1:** Generating Explanations

---

**Input**        : Sample image $x$, ML model $f(\cdot)$
**Output**      : Generated Conterfactuals $\tilde{\mathbf{x}}$

---

1  *Initialize the perturbations matrix parameter of size $n \times d$*
2  $\boldsymbol{\Sigma} \leftarrow randn(\mu = 0, \sigma = 0.01)$

3  *Get the original output from the ML model*
4  $y \leftarrow f(x)$

5  *Extract the latent features of the original input*
6  $z \leftarrow q_\phi(x)$

7  *Obtain fisher information on $z$*
8  $f_z \leftarrow \boldsymbol{F}(z)$

9  *Obtain $k$ partitions using spectral clustering*
10  $\boldsymbol{P} \leftarrow SpectralClustering(f_z)$

11  *Initialize counter*
12  $i \leftarrow 0$

13  **while** $i < \tau$ **do**
14      **for** *each $\epsilon, p \in (\boldsymbol{\Sigma}, \boldsymbol{P})$* **do**
15          *Perturb the latent features*
16          $\tilde{\mathbf{x}} \leftarrow p_\theta(\mathbf{z} + \boldsymbol{\epsilon})$
17          *Pass the perturbed image through the ML model*
18          $\hat{y} \leftarrow f(\tilde{\mathbf{x}})$
19          *Learn to reconstruct $\hat{Y}$ from $Y$*
20          $\mathcal{L} \leftarrow$ compute Eq. 4
21          Update $\epsilon$ while masking a subset of the gradients
22          $\boldsymbol{\epsilon} \leftarrow \boldsymbol{\epsilon} + \frac{\partial \mathcal{L}}{\partial \epsilon} \cdot p$
23      **end**
24      Update counter
25      $i \leftarrow i + 1$
26  **end**

---

# I OUT-OF-DISTRIBUTION EXPERIMENT

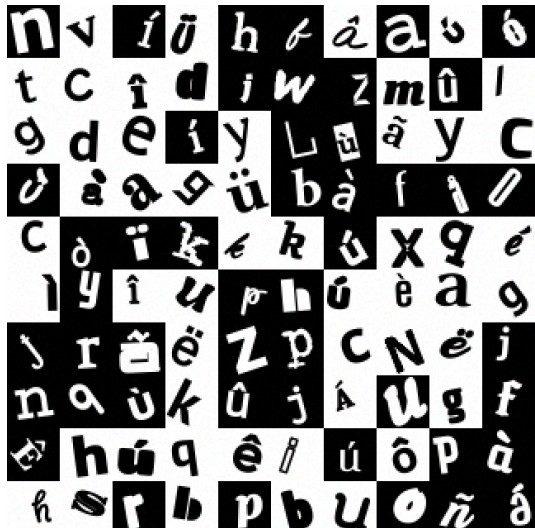

Figure 8: Sample of the synbols dataset.

We test the out-of-distribution (OOD) performance of DiVE with the Synbols dataset [31]. Synbols is an image generator with characters from the Unicode standard and the wide range of artistic fonts provided by the open font community. This provides us to better control on the features present in each set when compared to CelebA. We generate 100K black and white of 32×32 images from 48 characters in the latin alphabet and more than 1K fonts (Figure 8). In order to train the VAE on a different disjoint character set, we randomly select the following 32 training characters: {a, b, d, e, f, g, i, j, l, m, n, p, q, r, t, y, z, à, á, ã, å, è, é, ê, ë, î, ñ, ò, ö, ù, ú, û}. Counterfactuals are then generated for the remaining 16 characters: {c, h, k, o, s, u, v, w, x, â, ì, í, ï, ó, ô, ü}.

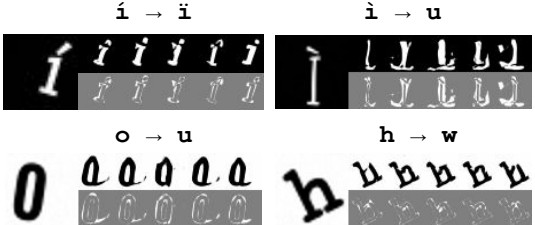

Figure 9: Successful counterfactuals for four different synbols in the OOD regime. Each sample consists of the original image in bigger size, five different counterfactuals generated by DiVE$_{\text{FisherSpectral}}$, and the difference in pixel space with respect to the original image (gray background). The header in each sample indicates the target class, *e.g.*, ï, u, w. All the counterfactuals are predicted by the ML model as belonging to the target class and differ from the oracle (*non-trivial*).

DiVE's objective is to discover biases on the ML model and the data. Thus, we use the *font* attribute in order to bias each of the characters on small disjoint subsets of fonts. Font subsets are chosen so that they are visually similar. In order to assess their similarity, we train a ResNet12 [40] to classify the fonts of the 100K images and calculate similarity in embedding space. Concretely, we use K-Means to obtain 16 clusters which are associated with each of the 16 characters used for counterfactual generation. The font assignments are reported in Table 7. Results for four different random counterfactuals are displayed in Figure 9. DiVE$_{\text{FisherSpectral}}$ successfuly confuses the ML model without changing the oracle prediction, revealing biases of the ML model.

| | |
|---|---|
| c | Anonymous Pro, Architects Daughter, BioRhyme Expanded, Bruno Ace, Bruno Ace SC, Cantarell, Eligible, Eligible Sans, Futurespore, Happy Monkey, Lexend Exa, Lexend Mega, Lexend Tera, Lexend Zetta, Michroma, Orbitron, Questrial, Revalia, Stint Ultra Expanded, Syncopate, Topeka, Turret Road |
| h | Acme, Alatsi, Alfa Slab One, Amaranth, Archivo Black, Atma, Baloo Bhai, Bevan, Black Ops One, Bowlby One SC, Bubblegum Sans, Bungee, Bungee Inline, Calistoga, Candal, CantoraOne, Carter One, Ceviche One, Changa One, Chau Philomene One, Chela One, Chivo, Concert One, Cookbook Title, Days One, Erica One, Francois One, Fredoka One, Fugaz One, Galindo, Gidugu, Gorditas, Gurajada, Halt, Haunting Spirits, Holtwood One SC, Imperial One, Jaldi, Jockey One, Jomhuria, Joti One, Kanit, Kavoon, Keania One, Kenia, Lalezar, Lemon, Lilita One, Londrina Solid, Luckiest Guy, Mitr, Monoton, Palanquin Dark, Passero One, Passion, Passion One, Patua One, Paytone One, Peace Sans, PoetsenOne, Power, Pridi, Printed Circuit Board, Rakkas, Rammetto One, Ranchers, Rowdies, Rubik Mono One, Rubik One, Russo One, Saira Stencil One, Secular One, Seymour One, Sigmar One, Skranji, Squada One, Suez One, Teko, USSR STENCIL, Ultra, Unity Titling, Unlock, Viafont, Wattauchimma, simplicity |
| k | Abel, Anaheim, Antic, Antic Slab, Armata, Barriecito, Bhavuka, Buda, Chilanka, Comfortaa, Comic Neue, Cutive, Cutive Mono, Delius, Delius Unicase, Didact Gothic, Duru Sans, Dustismo, Fauna One, Feronia, Flamenco, Gafata, Glass Antiqua, Gothic A1, Gotu, Handlee, IBM Plex Mono Light, IBM Plex Sans Condensed Light, IBM Plex Sans Light, Indie Flower, Josefin Sans Std, Kite One, Manjari, Metrophobic, Miriam Libre, News Cycle, Offside, Overlock, Overlock SC, Panefresco 250wt, Pavanam, Pontano Sans, Post No Bills Colombo Medium, Puritan, Quattrocento Sans, Quicksand, Ruda, Scope One, Snippet, Sulphur Point, Tajawal Light, Text Me One, Thabit, Unkempt, Varta, WeblySleek UI, Yaldevi Colombo Light |
| o | Abhaya Libre Medium, Abyssinica SIL, Adamina, Alice, Alike, Almendra SC, Amethysta, Andada, Aquifer, Arapey, Asar, Average, Baskervville, Brawler, Cambo, Della Respira, Donegal One, Esteban, Fanwood Text, Fenix, Fjord, GFS Didot, Gabriela, Goudy Bookletter 1911, Habibi, HeadlandOne, IM FELL French Canon, IM FELL French Canon SC, Inika, Jomolhari, Karma, Kreon, Kurale, Lancelot, Ledger, Linden Hill, Linux Libertine Display, Lustria, Mate, Mate SC, Metamorphous, Milonga, Montaga, New Athena Unicode, OFL Sorts Mill Goudy TT, Ovo, PT Serif Caption, Petrona, Poly, Prociono, Quando, Rosarivo, Sawarabi Mincho, Sedan, Sedan SC, Theano Didot, Theano Modern, Theano Old Style, Trocchi, Trykker, Uchen |
| s | Abril Fatface, Almendra, Arbutus, Asset, Audiowide, BPdotsCondensed, BPdotsCondensedDiamond, Bigshot One, Blazium, Bowlby One, Brazier Flame, Broken Glass, Bungee Outline, Bungee Shade, Butcherman, CAT Kurier, Cabin Sketch, Catenary Stamp, Chango, Charmonman, Cherry Bomb, Cherry Cream Soda, Chonburi, Codystar, Coiny, Corben, Coustard, Creepster, CriminalHand, DIN Schablonierschrift Cracked, Devonshire, Diplomata, Diplomata SC, Dr Sugiyama, DrawveticaMini, Eater, Elsie, Emblema One, Faster One, Flavors, Fontdiner Swanky, Fredericka the Great, Frijole, Fruktur, Geostar, Geostar Fill, Goblin One, Gravitas One, Hanalei Fill, Irish Grover, Kabinett Fraktur, Katibeh, Knewave, Leckerli One, Lily Script One, Limelight, Linux Biolinum Outline, Linux Biolinum Shadow, Lucien Schoenschriftv CAT, Membra, Metal Mania, Miltonian Tattoo, Modak, Molle, Mortified Drip, Mrs Sheppards, Multivac, New Rocker, Niconne, Nosifer, Notable, Oleo Script, Open Flame, Overhaul, Paper Cuts 2, Peralta, Piedra, Plaster, Poller One, Porter Sans Block, Press Start 2P, Purple Purse, Racing Sans One, Remix, Ringling, Risaltyp, Ruslan Display, Rye, Sail, Sanidana, Sarina, Sarpanch, Schulze Werbekraft, Severely, Shojumaru, Shrikhand, Slackey, Smokum, Sniglet, Sonsie One, Sortefax, Spicy Rice, Stalin One, SudegnakNo2, Sun Dried, Tangerine, Thickhead, Tippa, Titan One, Tomorrow, Trade Winds, VT323, Vampiro One, Vast Shadow, Video, Vipond Angular, Wallpoet, Yesteryear, Zilla Slab Highlight |
| u | Alef, Alegreya Sans, Almarai, Archivo, Arimo, Arsenal, Arya, Assistant, Asul, Averia Libre, Averia Sans Libre, Be Vietnam, Belleza, Cambay, Comme, Cousine, DM Sans, Darker Grotesque, DejaVu Sans Mono, Dhyana, Expletus Sans, Hack, Istok Web, KoHo, Krub, Lato, Liberation Mono, Liberation Sans, Libre Franklin, Luna Sans, Marcellus, Martel Sans, Merriweather Sans, Monda, Mplus 1p, Mukta, Muli, Niramit, Noto Sans, Open Sans, Open Sans Hebrew, PT Sans, PT Sans Caption, Raleway, Rambla, Roboto Mono, Rosario, Rounded Mplus 1c, Sansation, Sarabun, Sarala, Scada, Sintony, Tajawal, Tenor Sans, Voces, Yantramanav |
| v | Alex Toth, Antar, Averia Gruesa Libre, Baloo Bhai 2, Bromine, Butterfly Kids, Caesar Dressing, Cagliostro, Capriola, Caveat, Caveat Brush, Chelsea Market, Chewy, Class Coder, Convincing Pirate, Copse, Counterproductive, Courgette, Courier Prime, Covered By Your Grace, Damion, Dear Old Dad, Dekko, Domestic Manners, Dosis, Erica Type, Erika Ormig, Espresso Dolce, Farsan, Finger Paint, Freckle Face, Fuckin Gwenhwyfar, Gloria Hallelujah, Gochi Hand, Grand Hotel, Halogen, IM FELL DW Pica, IM FELL DW Pica SC, IM FELL English SC, Itim, Janitor, Junior CAT, Just Another Hand, Just Me Again Down Here, Kalam, Kodchasan, Lacquer, Lorem Ipsum, Love Ya Like A Sister, Macondo, Mali, Mansalva, Margarine, Marmelad, Matias, Mogra, Mortified, Nunito, Objective, Oldenburg, Oregano, Pacifico, Pangolin, Patrick Hand, Patrick Hand SC, Pecita, Pianaforma, Pompiere, Rancho, Reenie Beanie, Rock Salt, Ruge Boogie, Sacramento, Salsa, Schoolbell, Sedgwick Ave, Sedgwick Ave Display, Shadows Into Light, Short Stack, SirinStencil, Sofadi One, Solway, Special Elite, Sriracha, Stalemate, Sue Ellen Francisco, Sunshiney, Supercomputer, Supermercado, Swanky and Moo Moo, Sweet Spots, Varela Round, Vibur, Walter Turncoat, Warnes, Wellfleet, Yellowtail, Zeyada |
| w | Alata, Alte DIN 1451 Mittelschrift, Alte DIN 1451 Mittelschrift gepraegt, Amble, Arvo, Asap, Asap VF Beta, Athiti, Atomic Age, B612, B612 Mono, Barlow, Barlow Semi Condensed, Basic, Blinker, Bree Serif, Cairo, Cello Sans, Chakra Petch, Changa Medium, Cherry Swash, Crete Round, DejaVu Sans, Doppio One, Droid Sans, Elaine Sans, Encode Sans, Encode Sans Condensed, Encode Sans Expanded, Encode Sans Semi Condensed, Encode Sans Wide, Exo, Exo 2, Federo, Fira Code, Fira Sans, Fira Sans Condensed, Gontserrat, HammersmithOne, Hand Drawn Shapes, Harmattan, Heebo, Hepta Slab, Hind, Hind Kochi, IBM Plex Mono, IBM Plex Mono Medium, IBM Plex Sans, IBM Plex Sans Condensed, Iceland, Josefin Sans, Krona One, Livvic, Maak, Magra, Maven Pro, Maven Pro VF Beta, Mirza, Montserrat, Montserrat Alternates, Myanmar Khyay, NATS, Nobile, Oxanium, Play, Poppins, Prompt, Prosto One, Proza Libre, Quantico, Red Hat Text, Reem Kufi, Renner*, Righteous, Saira, Saira SemiCondensed, Semi, Share, Signika, Soniano Sans Unicode, Source Code Pro, Source Sans Pro, Spartan, Viga, Yatra One, Zilla Slab |
| x | Abhaya Libre, Amita, Antic Didone, Aref Ruqaa, Arima Madurai, Bellefair, CAT Sallust, CAT Linz, Cardo, Caudex, Cinzel, Cormorant, Cormorant Garamond, Cormorant SC, Cormorant Unicase, Cormorant Upright, Cuprum, Domine, Dustismo Roman, El Messiri, Fahkwang, FogtwoNo5, Forum, Galatia SIL, Gayathri, Gilda Display, Glegoo, GlukMixer, Gputeks, Griffy, Gupter, IBM Plex Serif Light, Inria Serif, Italiana, Judson, Junge, Libre Caslon Display, Linux Biolinum Capitals, Linux Biolinum Slanted, Linux Libertine Capitals, Lobster Two, Marcellus SC, Martel, Merienda, Modern Antiqua, Montserrat Subrayada, Mountains of Christmas, Mystery Quest, Old Standard TT, Playfair Display, Playfair Display SC, Portmanteau, Prata, Pretzel, Prida36, Quattrocento, Resagnicto, Risque, Rufina, Spectral SC, Trirong, Viaoda Libre, Wes, kawoszeh, okolaks |
| â | Accuratist, Advent Pro, Archivo Narrow, Aubrey, Cabin Condensed, Convergence, Encode Sans Compressed, Farro, Fira Sans Extra Condensed, Galdeano, Gemunu Libre, Gemunu Libre Light, Geo, Gudea, Homenaje, Iceberg, Liberty Sans, Mohave, Nova Cut, Nova Flat, Nova Oval, Nova Round, Nova Slim, NovaMono, Open Sans Condensed, Open Sans Hebrew Condensed, PT Sans Narrow, Port Lligat Sans, Port Lligat Slab, Pragati Narrow, Rajdhani, Rationale, Roboto Condensed, Ropa Sans, Saira Condensed, Saira ExtraCondensed, Share Tech, Share Tech Mono, Strait, Strong, Tauri, Ubuntu Condensed, Voltaire, Yaldevi Colombo, Yaldevi Colombo Medium |
| ì | Aladin, Alegre Sans, Allan, Amarante, Anton, Antonio, Asap Condensed, Astloch, At Sign, Bad Script, Bahiana, Bahianita, Bangers, Barloesius Schrift, Barlow Condensed, Barrio, Bebas Neue, BenchNine, Berlin Email Serif, Berlin Email Serif Shadow, Berolina, Bertholdr Mainzer Fraktur, Biedermeier Kursiv, Big Shoulders Display, Big Shoulders Text, Bigelow Rules, Bimbo JVE, Bonbon, Boogaloo, CAT FrankenDeutsch, CAT Liebing Gotisch, Calligraserif, Casa Sans, Chicle, Combo, Contrail One, Crushed, DN Titling, Dagerotypos, Denk One, Digital Numbers, Dorsa, Economica, Eleventh Square, Engagement, Euphoria Script, Ewert, Fette Mikado, Fjalla One, Flubby, Friedolin, Galada, Germania One, Gianna, Graduate, Hanalei, Jacques Francois Shadow, Jena Gotisch, Jolly Lodger, Julee, Kanzler, Kavivanar, Kazmann Sans, Kelly Slab, Khand, Kotta One, Kranky, Lemonada, Loved by the King, Maiden Orange, Marck Script, MedievalSharp, Medula One, Merienda One, Miltonian, Mouse Memoirs, Nova Script, Odibee Sans, Oswald, Paprika, Pathway Gothic One, Penguin Attack, Pirata One, Pommern Gotisch, Post No Bills Colombo, Princess Sofia, Redressed, Ribeye Marrow, Rum Raisin, Sancreek, Saniretro, Sanitechtro, Sevillana, Six Caps, Slim Jim, Smythe, Sofia, Sportrop, Staatliches, Stint Ultra Condensed, Tillana, Tulpen One, Underdog, Unica One, UnifrakturMaguntia, Yanone Kaffeesatz |
| í | Amatic SC, Bellota, Bellota Text, Bernardo Moda, Blokletters Balpen, Blokletters Potlood, Bubbler One, Bungee Hairline, Coming Soon, Crafty Girls, Gemunu Libre ExtraLight, Give You Glory, Gold Plated, Gruppo, IBM Plex Mono ExtraLight, IBM Plex Mono Thin, IBM Plex Sans Condensed ExtraLight, IBM Plex Sans Condensed Thin, IBM Plex Serif ExtraLight, IBM Plex Serif Thin, Julius Sans One, Jura, Lazenby Computer, Life Savers, Londrina Outline, Londrina Shadow, Mada ExtraLight, Mada Light, Major Mono Display, Megrim, Nixie One, Northampton, Over the Rainbow, Panefresco 1wt, Poiret One, Post No Bills Colombo Light, Raleway Dots, RawengulkSans, Sansation Light, Shadows Into Light Two, Sierra Nevada Road, Slimamif, Snowburst One, Tajawal ExtraLight, Terminal Dosis, Thasadith, The Girl Next Door, Thin Pencil Handwriting, Vibes, Waiting for the Sunrise, Wire One, Yaldevi Colombo ExtraLight |
| ï | Amiri, Buenard, Caladea, Charis SIL, Crimson Text, DejaVu Serif, Dita Sweet, Droid Serif, EB Garamond, Eagle Lake, Fondamento, Frank Ruhl Libre, Gelasio, Gentium Basic, Gentium Book Basic, Ibarra Real Nova, Junicode, Liberation Serif, Libre Baskerville, Libre Caslon Text, Linux Biolinum, Linux Libertine, Linux Libertine Slanted, Lusitana, Manuale, Merriweather, Noticia Text, PT Serif, Scheherazade, Spectral, Taviraj, Unna, Vesper Devanagari Libre |
| ó | ABeeZee, Actor, Aldrich, Alegreya Sans SC, Aleo, Amiko, Andika, Annie Use Your Telescope, Average Sans, Bai Jamjuree, Baumans, Belgrano, BioRhyme, Biryani, Cabin, Cabin VF Beta, Calling Code, Carme, Carrois Gothic, Carrois Gothic SC, Catamaran, Changa Light, Convincing, Droid Sans Mono, Electrolize, Englebert, Fresca, GFS Neohellenic, Imprima, Inconsolata, Inder, Inria Sans, Josefin Slab, K2D, Karla, Kulim Park, Lekton, Lexend Deca, Mako, McLaren, Meera Inimai, Molengo, Numans, Orienta, Overpass, Overpass Mono, Oxygen Mono, PT Mono, Panefresco 400wt, Podkova, Podkova VF Beta, Red Hat Display, Rhodium Libre, Ruluko, Sanchez, Sani Trixie, Sawarabi Gothic, Sen, Shanti, Slabo 13px, Slabo 27px, Sometype Mono, Space Mono, Spinnaker, Tuffy, TuffyInfant, TuffyScript, Ubuntu Mono, Varela |
| ô | Aguafina Script, Akronim, Alex Brush, Arizonia, Beth Ellen, Bilbo, Brauspulver, Calligraffitti, Cedarville Cursive, Charm, Clicker Script, Condiment, Cookie, Dancing Script, Dawning of a New Day, Dynalight, Felipa, Great Vibes, Herr Von Muellerhoff, Homemade Apple, Italianno, Jim Nightshade, Kaushan Script, Kristi, La Belle Aurore, League Script, Meddon, Meie Script, Mervale Script, Miama, Miniver, Miss Fajardose, Monsieur La Doulaise, Montez, Mr Bedfort, Mr Dafoe, Mr De Haviland, Mrs Saint Delafield, Norican, Nothing You Could Do, Parisienne, Petit Formal Script, Pinyon Script, Playball, Promocyja, Quintessential, Qwigley, Rochester, Romanesco, Rouge Script, Ruthie, Satisfy, Seaweed Script, Srisakdi, Vengeance |
| ü | Aclonica, Alegreya, Alegreya SC, Andada SC, Artifika, Averia Serif Libre, Balthazar, Bentham, Bitter, Cantata One, Crimson Pro, Croissant One, DM Serif Display, Eczar, Emilys Candy, Enriqueta, Faustina, Federant, Girassol, Grenze, Halant, Henny Penny, Hermeneus One, IBM Plex Serif, IBM Plex Serif Medium, IM FELL Double Pica, IM FELL Double Pica SC, IM FELL English, IM FELL Great Primer SC, Inknut Antiqua, Jacques Francois, Judges, Kameron, Laila, Lateef, Literata, Lora, Maitree, Markazi Text, Marko One, Monteiro Lobato, Original Surfer, Psicopatologia de la Vida Cotidiana, Radley, Rasa, Ribeye, Roboto Slab, Rokkitt, Rozha One, Sahitya, Sansita, Simonetta, Source Serif Pro, Spirax, Stardos Stencil, Stoke, Sumana, Uncial Antiqua, Vidaloka, Volkhov, Vollkorn, Vollkorn SC |

Table 7: Font clusters assigned to each character.

## J   Details on the Bias Detection Metric

In Table 1, we follow the procedure in first developped in [26] and adapted in [53] and report a confounding metric for bias detection. Namely, the "Male" and "Female" is the accuracy of the oracle on those class conditioned on the target label of the original image. For example, we can see that the generated explanations for the the biased classifier, most methods generated an higher amount of Non-smiling females and smiling males, which was expected. The confounding metric, denoted as overall, is the fraction of generated explanations for which the gender was changed with respect to the original image. It thus reflect the magnitude of the the bias as approximated by the explainers. Singla et al. [53] consider that a model is better than another if the confounding metric is the highest on $f_{\text{biased}}$ and the lowest on $f_{\text{unbiased}}$.

This is however not entirely true. There is no guarantee that $f_{\text{biased}}$ will perfectly latch on the spurious correlation. In that case, an explainer's ratio could potentially be too high which would reflect an overestimation of the bias. We thus need to a way to quantify the gender bias in each model. To do so, we look at the difference between the classifiers accuracy on "Smiling" when the image is of a "Male" versus a "Female". Intuitively, the magnitude of this difference approximates how much the classifier latched on the "Male" attribute to make its smiling predictions. We compute the same metric for in the non-smiling case. We average both of them, which we refer as ground truth in Table 1. As expected, this value is high for the $f_{\text{biased}}$ and low for $f_{\text{unbiased}}$. Formally, the ground truth is computed as

$$\mathbb{E}_{a \sim p(a)} \Big[ \mathbb{E}_{x,y \sim p(x,y|a)} \big[ \big| \mathbb{1}[y = f(x)|a = a_1] - \mathbb{1}[y = f(x)|a = a_2] \big| \big] \Big] \tag{10}$$

where $a$ represents the attribute, in this case the gender.

