# OpenReview forum: "Beyond Trivial Counterfactual Generations with Diverse Valuable Explanations"
_ICLR.cc/2021/Conference — Reject_

### Official Review · AnonReviewer3 · 2020-10-19
**An interesting work not entirely validated and with missing references**

**Rating:** 4
**Confidence:** 4

**Review:**

The present work proposes an explanation method returning actionable, proximal, diverse, and not trivial counterexamples as explanation. The work is well written even though various concepts are detailed only in the Appendix.
The proposal is interesting, sound in the formulation, and valuable from the experiments reported. The examples reported are nice and quite convincing. The bias detection case study is effective and well presented.
However, the paper lacks some major points that make it not ready for publication.
First, even though theoretically the proposed DIVE method can be employed on any type of data it is developed and tested only on image data. The future work discussion is missing and the possibility to employ it on other data types is not treated. The fact that experiments are reported for a unique dataset is a limitation. Various simple datasets with more simple features can be adopted (mnist, cifar10, fashion mnist) or cifar100 imagenet by considering categories and specific classes for the different features.
Second, the paper misses various work on counterfactual explanations (some of them are listed in the following) and consequently also a comparison against them. In particular, it would be interesting to test DIVE against methods using a different logic for finding the counterexamples than against methods using a similar approach. Furthermore, in DICE (cited) are presented many evaluation measures not adopted in this paper and it is not justified why.
Finally, the trivial/valuable explanations, which are the main motivation for this wor, are not formally defined in Section 4, nor in Section 5.

Minor issues:
- The optimization problem solved by DIVE to find the exemplars is not sufficiently detailed in the main paper.
- It is not clear (or not easy to find) the dimension of the latent space of the VAE and how it affects the performance

Missing Related
Karimi, Amir-Hossein, et al. "Model-agnostic counterfactual explanations for consequential decisions." International Conference on Artificial Intelligence and Statistics. 2020.
Poyiadzi, Rafael, et al. "FACE: feasible and actionable counterfactual explanations." Proceedings of the AAAI/ACM Conference on AI, Ethics, and Society. 2020.
Guidotti, R., Monreale, A., Matwin, S., & Pedreschi, D. (2019, September). Black Box Explanation by Learning Image Exemplars in the Latent Feature Space. In Joint European Conference on Machine Learning and Knowledge Discovery in Databases (pp. 189-205). Springer, Cham.
Pawelczyk, M., Broelemann, K., & Kasneci, G. (2020, April). Learning Model-Agnostic Counterfactual Explanations for Tabular Data. In Proceedings of The Web Conference 2020 (pp. 3126-3132).
Dhurandhar, A., Chen, P. Y., Luss, R., Tu, C. C., Ting, P., Shanmugam, K., & Das, P. (2018). Explanations based on the missing: Towards contrastive explanations with pertinent negatives. In Advances in Neural Information Processing Systems (pp. 592-603).
Van Looveren, A., & Klaise, J. (2019). Interpretable counterfactual explanations guided by prototypes. arXiv preprint arXiv:1907.02584.
Guidotti, R., Monreale, A., Giannotti, F., Pedreschi, D., Ruggieri, S., & Turini, F. (2019). Factual and counterfactual explanations for black box decision making. IEEE Intelligent Systems, 34(6), 14-23.

---

> ### Author Response · Authors · 2020-11-25
> **Response to Reviewer 3**
>
> Thank you for the helpful comments! We address each of them below. You can find questions shared with other reviewers are answered in the “general response”.
>
> **Could you include a future work section discussing the possibility to employ it on other data types?**
>
> Yes, we have included them please see our response above with the heading "General Response".
>
> **Could you adopt a dataset with simpler features?.**
>
> Yes, we have included experiments on Synbols (Lacoste et al. 2020). See our general response for details.
>
> *Lacoste, Alexandre, et al. "Synbols: Probing learning algorithms with synthetic datasets." Advances in Neural Information Processing Systems 33 (2020).*
>
>
> **In DICE (cited) are presented many evaluation measures not adopted in this paper and it is not justified why.**
>
> We have extended the text with more details comparing DiVE and DICE. DICE evaluates its set of CFs on validity, proximity, and sparsity. The first measure is taken into account by our method (we report the ratio of valid CFs). For proximity, we provide multiple metrics such as the FID (Table 2), latent space closeness and face verification accuracy (Table 4 in Appendix C), and VGG_Face embedding similarity (Figure 2). For sparsity, we have included new results (Table 3) with the average attribute change (as suggested by DICE). We have also extended the comparison with DICE in the related work. We clarify that DICE directly perturbes the observed features and does not aim to find non-trivial explanations.
>
> **Could you define trivial/valuable explanations?**
>
> Yes, we have defined them please see our response above with the heading "General Response".
>
> **The optimization problem solved by DIVE to find the exemplars is not sufficiently detailed in the main paper.**
>
> The optimization problem consists of minimizing Eq. 4 by gradient descent. We have updated Section 3 with this information. In addition, we have included an extra reference to Algorithm 1 in the Appendix and improved its description.
>
> **What is the dimension of the latent space of the VAE and how it affects the performance?**
>
> The ELBO has a natural principled way of selecting the dimensionality of the latent representation. If d is larger than necessary, it will not enhance the reconstruction error and the optimization of the ELBO will make the posterior equal to the prior for these extra dimensions. More can be found on the topic in (Lucas et al. 2019). In practice, we experimented with d={64, 128, and 256} and found that with d=128 we achieved a slightly lower ELBO. This was reported in Appendix D, and we have updated the text including more information.
>
> *Lucas, James, et al. "Understanding posterior collapse in generative latent variable models." (2019).*
>
> **There is missing related work.**
>
> Thanks, we have included all the suggested references and restructured the related work (see general response).

---

### Official Review · AnonReviewer2 · 2020-10-28
**Official Blind Review #2**

**Rating:** 4
**Confidence:** 4

**Review:**

## Reasons for score

Overall, I really liked your proposed method and would appreciate seeing your paper published. However, as of now, it does not pass the acceptance threshold. If you address my questions and requests, I would be willing to change my score.
I think that it is critical that you improve the experimental section in terms of writing and presentation of the experimental protocol, metrics, results and especially section 4.3. Also, I think that it is necessary that you improve the related work section and highlight the limitations of your method.


## My background

My research is focused on detecting data biases (or spurious correlations) learned by deep neural networks using explainability methods. This is the exact scope of this paper. However, although I have a solid understanding of the attribution methods, this paper develops a counterfactual one which is related to but not directly within my area of expertise.

## Summary

Context:
The paper focuses on counterfactual explainability methods that aim at improving the reliability of machine-learning systems and help for model debugging (finding spurious correlations or model biases). Given an input example and a target prediction score, this class of methods generates counterfactual examples.

Problem:
The authors identify and tackle issues of state-of-the-art methods, xGEM[23] and PE[41]:
- they combine multiple biases of the model in a single counterfactual example (not disentangled),
- they exaggerate or remove the presence of the attribute being classified (trivial and not valuable).

Solution and novelty:
The proposed method generates $n$ counterfactual examples for a given input example and a target prediction score. It is composed of :
- a pretrained encoder-decoder architecture with $\beta$-TCVAE[5] that produces a disentangled vectorial representation of the input example,
- an algorithm that produces $n$ perturbations of this representation that are decoded to generate counterfactual examples.
The perturbations are obtained by minimizing a loss composed of:
- a binary cross-entropy loss to generate examples that match the target prediction score,
- an L1 loss between the input example and each generated example to force small perturbations in input space,
- an L1 loss on each perturbation to force small perturbations in latent space,
- a structural mechanism based on the Fisher information matrix and spectral clustering to force diversity of perturbations

Claim:
The proposed method generates counterfactuals that are diverse, non-trivial (i.e. not just exaggerate or remove an attribute), high quality (i.e. in distribution) and valuable explanations about the model's prediction (i.e. biases can be detected by humans).

Experiments:
The method reaches state-of-the-art results on two existing benchmarks. The paper also introduces a new benchmark to evaluate how valuable the explanations are.

## What I liked the most

- meta-problem of explaining neural networks is critical
- mostly well contextualized
- mostly easy to read
- mostly easy to understand
- mostly well illustrated
- relevant issues have been identified
- novel, simple and interesting method to tackle them
- novel experimental benchmark
- I really liked the hat section of 4. Experimental Results where you describe the 3 different aspects that you evaluate and you point to the associated sections
- improvements over state-of-the-art are significant
- ablation study mostly validates each proposed components


## What could be improved

My cons are expressed per section, but the listing is random (I did not write the critical cons at the top of each section).

1. Abstract and introduction
- I find it surprising to assume derivability for a black box in the context of explainability methods. I know PE[41] uses this definition, but at least they clearly state that they assume derivability. For me, it does not correspond to the commonly admitted definition of a black box (e.g. Wikipedia: "implementation is opaque" "without any knowledge of its internal workings"). See the influential Ribeiro et al. 2016 "Explaining the predictions of any classifier" (4000 citations) for a definition that implies no assumption about derivability (i.e. "model-agnostic").
- I had to write a detailed summary of your paper to better understand it. It is not critical, but I think that your writing can be further improved/structured to better frame the issues of previous state-of-the-art that you tackle, the novelty of your contributions (for instance, is it novel to structure diversity with FisherMatrix and spectral clustering?) and the claims that you validate in the experiments.


2. Related work
- First of all, note that reading the extended related work in the supplementary material (which is not mandatory) did not address the points I am going to make. I am convinced that your related work section could be improved as follows: 1) I would briefly mention that many post-hoc explainability approaches to detect biases exist and that you focus on counterfactual ones. 2) I would write about counterfactual methods that are not generative (see Papernot et al. 2018 "Deep k-Nearest Neighbors: Towards Confident, Interpretable and Robust Deep Learning" which may be related). 3) I would write about state-of-the-art generative counterfactual methods and their limitations. 4) I would write about explainability methods that focus on diversity (you should mention that only a few exist if it is the case).
- I do not agree with "[attribution methods] do not explain how to modify [input features] to change the model outcome". I think that is exactly what they do (by applying perturbations/masks). See Fong et al. 2017 "Interpretable explanations of black boxes by meaningful perturbation" that you already cite. However counterfactual generative methods of your kind apply perturbations that look real from a human standpoint or "in-distribution". I would like to see a discussion about the need for this kind of perturbations.

3. Proposed Method
- From a first read, it is not easy to map Dive, DiveFisher and DiveFisherSpactral onto their definition (around Eq. 7 and 8). Paragraphs Diversity loss and Beyond trivial counterfactual explanations could be better structured in this respect.
- Figure1: You should make it clear that counterfactuals at the top are non-valuable to detect biases (because considered not bald by humans and by the model), whereas the bottom are valuable to detect biases (because considered bald by humans but not by the model).
- Figure1: I do not understand why the bottom right has a black border.

4. Experimental results
- Overall, I do not find it clear what your training, validation and testing sets are made of. Do you use a testing set? How do you tune your hyperparameters? Is it standard?
4.1
- What does FID stand for?
- I do not understand this: "we train a second-order spline on the trajectory of perturbations produced during the gradient descent steps of our method"
- I think I understand, but it could be more clearly stated: "even though DiVe is not explicitly trained to produce examples at intermediate target probabilities". Actually, you can choose the target probability, isn't it?
4.2
- Table2: metric is not included in the caption
- Table2: I do not understand the reason why some numbers are bold
4.3
- Overall I think it is critical to improving this section. I did not clearly understand how your novel experimental protocol can be used to validate "the ability to identify diverse valuable explanations".
-"Because it is costly to provide a human evaluation of an automatic benchmark, we approximate both the proximity and the real class with the VGGFace2-based oracle." Why do you think that we can trust the approximation?
- Why don't you compare against PE and xGEM even tho it's not the same decoder as xGEM+ and DiVE?
- I would make it clear what "success rate" means (you could make it bold where you define it)
- Figure3: Overall, I should be able to mostly understand the Figure by reading the caption. It is not the case at all. I did not find it clear that you explore different hyperparameters.
- "We show results for all explanations in Figure 3a and only when the generated images are counterfactuals in Figure 3b." It is not clear. From what I understood your method generates counterfactuals only! Some of them are just valuable to detect biases (i.e. misclassified by humans) or not. What does "successful counterfactuals" mean?
- It would be interesting to have a baseline with random masks to compare against your proposed method (Fisher and FisherSpectral).
- There are no experiments on Dive-Fisher or Dive-FisherSpectral except for Figure 3 (and even for Figure 3. You don’t explain the difference between results for the different versions of Dive). I did not understand which version of your method works best. Also, you should define Dive-F and Dive-FS or not use them. I find that using Dive_Fisher and Dive_FisherSpectral improves clarity.

5. Conclusion section
- I would like to see a discussion about the limitation of your method and issues that need to be addressed in future works. In particular, what happens if your encoder-decoder is biased? What happens when your encoder-decoder is not able to produce disentangled representations (it seems often the case on real and complex datasets)? How to calibrate the many hyperparameters (to train the encoder-decoder and to weigh your loss functions)?

---

> ### Author Response · Authors · 2020-11-25
> **Response to Reviewer 2 (2/2)**
>
> **Table1: I do not understand the reason why some numbers are bold**
>
> In a biased classifier on gender where all males are smiling and all females are not smiling, the generated counterfactuals for the target “smile” should contain mostly men. As indicated by Singla et al., 2020, we can estimate the bias of a classifier by counting the number of counterfactuals that have switched gender to explain the attribute “smiling”. Thus, the more counterfactuals that switch the gender, the better is the explainer at discovering the gender bias of the classifier. For the unbiased classifier, none of the counterfactuals should contain a change of gender, and thus, lower gender switch values indicate that the explainer is better at discovering the gender bias for the attribute “smiling”. Thus, in PE, they mark in bold the highest and the lowest values for the biased and the unbiased classifiers respectively. Instead of the highest and the lowest gender switch values, we propose to mark in bold those which are closest to the real amount of bias of the classifier, which we denote as “ground truth” in Table 1.
>
> **I did not clearly understand how your novel experimental protocol can be used to validate "the ability to identify diverse valuable explanations".**
>
> In the updated manuscript, we have clarified that “valuable” means “valid”, “non-trivial”, and “proximal” (including “sparse”). Thus in Figure 2, we display the ratio of counterfactuals that are “valid” and “non-trivial” in the Y axis, and the proximity in embedding space in the X axis. Non-trivial examples are those for which an oracle predicts a different class than the ML model. Diversity is implicitly measured since the main difference between “DiVE” and “xGEM+” is the diversity loss.
>
> **Why do you think that we can trust the oracle approximation?**
>
> We cannot trust it but we have included a human evaluation reaching similar results. See the general response for more details.
>
> **Why don't you compare against PE and xGEM even though it's not the same decoder as xGEM+ and DiVE?**
>
> PE must be trained from scratch for each CelebA class and the authors did not provide the code nor the models for other classes rather than Young, and Bangs. Moreover, we have not been able to reproduce the results for the attribute Bangs. However, we have been able to introduce the results for “Young” in Table 3 in the updated manuscript. Results indicate that DiVE is more successful than PE at finding non-trivial counterfactuals.
>
> **I would make it clear what "success rate" means (you could make it bold where you define it)**
>
> Success rate is the ratio of valid explanations that are non-trivial, we call these counterfactuals “successful counterfactuals”. We have updated the text and put the term in italic.
>
> **Could you improve the caption of Figure2?**
>
> Yes, we have extended the caption explaining that the points and the curves correspond to a hyperparameter sweep.
>
> **"We show results for all explanations in Figure 3a (Initial submission) and only when the generated images are counterfactuals in Figure 2b." It is not clear. From what I understood your method generates counterfactuals only! Some of them are just valuable to detect biases (i.e. misclassified by humans) or not.**
>
> Note that it is not always possible to find a counterfactual that fools the ML model while keeping the proximity and sparsity constraints (nor do PE or xGEM). In Figure 2, a “successful counterfactual” is “valid” and “non-trivial”. Figure 3a (initial submission) does also take into account non-valid counterfactuals, which decreases the number of successful counterfactuals of all the methods. We have decided to remove Figure 3a (in the initial submission), since it did not provide any additional insights and we have replaced it with an OOD experiment on Synbols.
>
> **It would be interesting to have a baseline with random masks to compare against your proposed method (Fisher and FisherSpectral).**
>
> We have included random masks in Figure 2. We observed that they are less successful than using Fisher-based masks at finding non-trivial counterfactuals.
> Dive-Fisher or Dive-FisherSpectral only appear in Figure 2 as Dive-F and Dive-FS and the difference in results is not explained.
> We have introduced a new Table 3 to compare the different versions of DiVE with PE and xGEM. We have also replaced DiVE-F and DiVE-FS by DiVE_Fisher and DiVE_FisherSpectral and improved the description of the results.
>
> **Could you include discussion, limitations, and future work?**
>
> Yes, see our general response.

---

> ### Author Response · Authors · 2020-11-25
> **Response to Reviewer 2 (1/2)**
>
> Thanks for your thorough and constructive review! Below we address each of your comments. Questions shared with other reviewers are answered in the “general response”.
>
> **A black box should not be derivable.**
> We agree that it could be misleading and we have changed “black-box” by “ML model”. What we meant by black box is that the model is not-interpretable. Our explainer is model agnostic and only requires access to the ML model gradients.
>
> **Could you improve/structure better the previous state-of-the-art, the novelty of the contributions, and the claims validated in the experiments?**
>
> Yes. We have made a great effort to improve the introduction and clarify the contributions with respect to previous state of the art. In the updated introduction, we first introduce the concepts of actionable, sparse, valid, proximal, and diverse in the third paragraph and link them to the literature. In the fourth paragraph, we explain the need for non-trivial explanations. In the fifth paragraph we introduce DiVE. Finally, in the sixth paragraph, we relate all the introduced properties with their respective experiments.
>
> **Could you restructure the related work?**
>
> Yes, we have restructured and extended the related work accordingly.
>
> **I would like to see a discussion about the need for this in-distribution perturbations.**
>
> Photo-realistic (or in-distribution) generated explanations satisfy three key criteria for counterfactual explanations: they are proximal to the input, they contain sparse changes, and they are actionable in the sense that they propose plausible changes to the input. Methods that act on pixel space tend to produce visibly noisy patterns. So while they provide informative counterfactuals, they fail to satisfy the plausibility criterion. This is particularly important in use-cases like imaging diagnostics for radiology, or pathology.
>
> **Could you improve the map of Dive, DiveFisher and DiveFisherSpectral onto their definition?**
>
> Yes. We have updated sections 3.2 and 4 including the definitions of the different variants of DiVE.
>
> **Could you make clear that counterfactuals at the top of Figure1 are non-valuable whereas the bottom are valuable? I do not understand why the bottom right has a black border.**
>
> We have updated Figure 1 and its description in Section 4.3, explaining that the top examples are trivial, while the bottom examples are non-trivial. We use “trivial” instead of “valuable” since it is the critical difference between the two rows.
>
> **What data spits do you use and how do you tune hyperparameters?**
>
> For fair comparison we used the same experimental setup as the one from progressive exaggeration (Singla et al. 2020). See the general comments for details.
>
> **What does FID stand for?**
>
> FID is the Frechlet Inception Distance, a standard metric to measure the quality of generated images. It was originally proposed by (Martin et al. 2017). First a pre-trained Inception v3 model is used to generate feature vectors for a set of real images and a set of generated images, then the distance between the two sets is computed using the mean and covariance matrix of their respective feature vectors (Martin et al. 2017). The more similar the two sets are the smaller the FID.
>
> *Heusel, Martin, et al. "Gans trained by a two time-scale update rule converge to a local nash equilibrium." Advances in neural information processing systems. 2017.*
>
> **I do not understand this: "we train a second-order spline on the trajectory of perturbations produced during the gradient descent steps of our method"**
>
> The model in progressive exaggeration is built so that it can generate explanations that fool the classifier for arbitrary target confidence values. Our method is designed to maximize the confidence on the wrong class through gradient descent and thus we needed a way to obtain counterfactuals at intermediate target confidence values. During the gradient descent of DiVE we obtain some intermediate confidence values but they do not match the target confidences. However, we can use them and their corresponding latent vectors to fit a function that tells us what the latent vector should be for an arbitrary target confidence. In our case we chose to use a piecewise quadratic polynomial, which is typically called Spline in the computer graphics literature. We have included this explanation in the updated version of the submission.
>
> **You can choose the target probability, isn't it?**
>
> Yes. However, that would mean to train DiVE for a significantly longer time (until each of the targets is reached), and we found that the Spline was already more precise achieving the target probabilities than the method proposed in PE.
>
> **Table1: metric is not included in the caption**
>
> The metric is the ratio of counterfactuals classified as the target class. We have updated the caption.

---

### Official Review · AnonReviewer4 · 2020-10-28
**A method that counterfactually explains predictions of classifiers by exploiting perturbations on data samples.**

**Rating:** 7
**Confidence:** 3

**Review:**




In this paper, the authors present a method based on counterfactuals that learns a perturbation using constraints to ensure diversity in explanations.

The authors argue that explanations produced by their method are more “actionable, diverse, valuable and proximal than the previous literature”. However,
it is unclear how they quantitatively measure these attributes, given that FID scores only captures the similarity of generated images to real ones.

I would like to understand the motivation on using the perceptual reconstruction loss. The authors should clarify the usage of this loss in their method and
highlight its importance on their explanatory method. The author briefly mentioned the gains in terms of image quality, when compared with GANs in PE.
However, I would like to see a more deeper discussion.

Since interpretability is closely related to users/humans, it is difficult to assess the quality of the generated explanations without human evaluations.
An initial setup could be the one used in PE.

Overall, assuming the above limitations, the experiments help to understand the contributions of the article.

Typos:

- Sec. 3.3: “Since these mask are …” -> “Since these masks are…”
- Sec. 4.2: “In Figure 2b, …” -> “In Figure 2, …”

---

> ### Author Response · Authors · 2020-11-25
> **Response to Reviewer 4**
>
> Thanks for the positive feedback! We answer your questions below. Questions shared with other reviewers are answered in the “general response”.
>
> **Besides FID, it is unclear how the “actionable, diverse, valuable and proximal” properties are evaluated.**
>
> In addition to FID, we also compare with xGEM and PE in face verification and latent closeness scores, see Table 4 in  Appendix C. In Figure 3 we also report the latent space closeness of a VGG-Face2 model. In this new version of the manuscript, we have included an additional table showing that the explanations produced by DiVE change less attributes on average (they are more sparse) compared to previous literature.
>
> **What is the motivation and importance of the perceptual reconstruction loss?**
>
> One of the main reasons why previous methods such as PE use GANs is because plain VAEs tend to produce blurry images, which makes the explanations less proximal. Thus, based on previous literature (Hou et al. 2017), we decided to use a perceptual loss to obtain more proximal counterfactuals, and we reported results with and without the perceptual loss for a more complete comparison (DiVE vs DiVE--). We have included this clarification in the text.
>
> **Could you include a human evaluation?**
>
> Yes we have included it, please see our response above with the heading "General Response".

---

### Official Review · AnonReviewer1 · 2020-10-28
**interesting but execution needs substantial work**

**Rating:** 6
**Confidence:** 5

**Review:**

Summary: The authors propose interpreting the decision of a black-box (BB) image classifier using diverse counterfactual explanations. The proposed model consists of a pre-trained β-TCVAE, which learns to extract a disentangled latent representation for the input image. To generate explanations for a given image, the model optimizes to find n latent perturbations. Each decoded output from β-TCVAE is similar to the original image and produces a desired outcome from the BB classifier. To ensure the diversity among the n latent perturbations, the model minimizes the pairwise similarity loss between the latent perturbations. The model further performs spectral clustering to partition the latent space into different attributes. Thus, at inference time, for the same input image, multiple counterfactual images can be generated as explanations by changing different dimensions of the latent space. The experiments demonstrate the realistic quality of the explanations and their ability to discover bias in the BB classifier.


•	The idea of generating multiple images as counterfactual explanations is interesting. If multiple explanations differ in multiple attributes, it can help identify un-wanted correlation or biases in the model and datasets.
•	The paper lacks detailed experiments to quantify the importance of diverse explanations. The experiment should quantify what attributes, apart from trivial counterfactual changes, differ across the explanation images. The successful explanation experiment confirms differences in the explanation images, but ``what" is different is not apparent.
•	The author's definition of a valuable explanation is misleading. In the introduction, the authors describe a valuable explanation as an explanation that is proximal, i.e., it is much similar to the input image, and actionable i.e., it can be derived by performing feasible changes to the input image. In the experiment section on "Beyond trivial explanations," authors defined a valuable explanation as the one for which the BB model and human or its proxy (an oracle network) have different outcomes. For the two definitions to be consistent, the authors assume that the oracle network can uniquely identify feasible features in an image and consider such features in its classification decision.
•	Feasible features in an image are hard to define. For example, adding/removing sunglasses is a trivial example of a feasible change. While changing some pixels around the face's lips, to add/remove the smile is a more complex change whose feasibility is hard to define. As the change in the region around the lips may add/remove an expression from a face that is hard to quantify. Also, the authors didn't perform any experiments to show that the explanations generated by their method correspond to feasible changes in the input image, in contrast to other methods like PE or xGEM. Since all the methods involve a generative process, which is prone to perform un-realistic changes to the image, the author's claim of restricting perturbation to only feasible changes is vague.
•	It is not clear what training data is being used to train the encoder-decoder in the proposed model. If the method uses a different dataset from the training dataset of the BB classifier, please state that explicitly. Also, to compare against the existing methods, the authors can design an experiment where they consider a dataset (explain-dataset) different from the dataset used for training the BB classifier (BB-dataset). They can then use the explain-dataset to train and compare the different explanation models (DiVE, PE, xGEM).
•	The authors consider a perceptual reconstruction loss instead of a standard pixel-wise reconstruction. An ablation study to compare the different reconstruction losses is required to justify the proposed model's additional dependency on a pre-trained network R.
•	The term "adversarial loss" is misleading. The adversarial loss defined in equation 5 constrains the model to learn a perturbation that results in the desired probabilistic outcome. Hence, the name should reflect this constraint and its dependence on the BB classifier. As mentioned in the text, there are no adversaries here; hence no min-max game to be solved.
•	The authors claim that the sparsity constraint on the latent perturbation results in proximal and actionable explanations. The explanations' actionability is defined in terms of sparsity in the number of attributes that are modified in the latent perturbation. Since β-TCVAE is trained in an unsupervised manner, the latent space is not explicitly disentangled in measurable attributes (e.g. presence of sunglasses). Hence, the disentangled attributes learned by β-TCVAE, and discovered by the spectral clustering, may not correspond to discrete human-understandable concepts (e.g., sunglasses). An actionable explanation enables the human end-users to modify discrete concepts (e.g., remove sunglasses from the face) and observe changes in the BB's behavior. Experiments are required to quantify the actionability of the explanations.
•	A replication of the "Beyond trivial explanations" experiment on real images can also demonstrate the disagreement between the prediction from the BB classifier and the oracle classifier. It's not clear how experimenting on the counterfactual images provided any more/different information than the same experiment performed on real images.
•	A valid counterfactual image should produce an opposite prediction from the BB classifier compared to the input image There are no experiments to quantify the validity of counterfactual explanations.

---

> ### Author Response · Authors · 2020-11-25
> **Response to Reviewer 1 (2/2)**
>
> **Why do counterfactual images provide any more/different information than real images?**
>
> Because using real examples would limit explanations to cases existing in the training data. In the case of CelebA, this would involve having images for all the possible actionable attributes for each celebrity in the dataset for all the other possible actionable variations such as illumination. In contrast, a generative model that learns to disentangle some of these factors could be used to modify them in any image, even when no training data for that image is available.
>
> **There are no experiments to quantify the validity of counterfactual explanations.**
>
> Table 1 quantifies the ratio of valid counterfactuals. The rest of the experiments are performed on the valid counterfactuals.

---

> ### Author Response · Authors · 2020-11-25
> **Response to Reviewer 1 (1/2)**
>
> Thank you for the valuable feedback! We answer each of your questions below. Questions shared with other reviewers are answered in the “general response”.
>
> **Could your experiment quantify what attributes, apart from trivial counterfactual changes, differ across the explanation images?**
>
> Yes, we have included a new experiment that evaluates the sparsity of the counterfactual explanations, i.e., the number of attributes that a counterfactual explanation changes. Table 3 shows that DiVE (our approach) generates sparser explanations than xGEM+ and PE. Moreover, DiVE$_{FisherSpectral}$ is the method that produces the sparsest explanations among all the baselines. This is important since explanations that change fewer attributes remain closest to the input and tend to be more actionable.
>
> **The definition of “valuable” is different in the introduction and Section 4.3.**
>
> We have updated the definition as detailed in our general response.
>
> **The author's claim of restricting perturbation to only feasible changes is vague.**
>
> We agree we used “feasible” somewhat vaguely in the original manuscript, thank you. Accordingly, we now clearly define and evaluate with experiments each of the properties of the method, i.e., validity, proximity, sparsity, non-triviality. A method that provides a diverse set of proximal and sparse explanations is more likely to produce feasible explanations. Explanations that have high “proximity”, i.e., they are close to the input, and high “sparsity”, i.e., few attributes have been changed with respect to the input, are more likely to be feasible.
>
> To evaluate validity, we provide the ratio of valid counterfactuals for biased and unbiased classifiers in Table 1. To evaluate “proximity”, we report FID scores in Table 2 and face verification and latent space closeness scores in Table 3, showing that DiVE produces higher quality images which are better than PE and xGEM at keeping the original identity of the face. To evaluate “sparsity”, we provide a comparison in Table 3 showing that DiVE changes less attributes on average than PE and xGEM. Non-triviality is evaluated in Figure 2 and Table 3, 5, respectively, where we demonstrate that DiVE is more successful than PE and xGEM at finding non-trivial counterfactuals.
>
> **Could you explicitly state the training data for the VAE and the BB?**
>
> Yes, we follow the same procedure as PE (Singla et al. 2020), see our general response for more details.
>
> **Could you produce explanations on a different dataset?**
>
> Yes. We have included out-of-distribution experiments on the Synbols benchmark, see our general response for more details.
>
> **Could you include an ablation study comparing the perceptual loss with the pixel-wise reconstruction loss?**
>
> The manuscript already includes an experiment comparing DiVE (with the perceptual loss) with DiVE-- (with pixel-wise reconstruction loss) in Table 2. Moreover the new experiments on Synbols dataset do not use a perceptual loss and achieves competitive performance.  We have clarified this in the paper.
>
> **The term "adversarial loss" is misleading.**
>
> We have changed it to “counterfactual loss” to avoid possible confusions. Accordingly, we have also changed the “adversarial regularization loss” to “proximity loss”. Note that we drew the “adversarial loss” term from the “adversarial attack” literature where there are no min-max games.
>
> **How do you quantify actionability and how does it relate to sparsity, proximality, and the disentangled latents of the β-TCVAE, which may not learn human-understandable concepts?**
>
> We agree that the concepts of “proximity”, “sparsity” and “actionable” were not correctly defined in the original manuscript. We have made a significant effort to clarify all these terms in Section 1  in the new version of the manuscript and we provide experiments validating each of these properties. We state that a diverse set of valid, proximal, sparse, and non-trivial explanations is more likely to contain actionable explanations.
>
> To quantify the effect of disentanglement, we have trained a new version of xGEM+ with disentanglement (xGEM++) and report the results in Table 3. We found that disentangled representations provide the explainer with a more precise control on the factors being perturbed, which results in an increase the success rate of the explainer by 16\%.
>
> About the human-understandable concepts, we expect that progress in identifiable representations will bring us closer to human perception. Meanwhile, we combined the best tools in the literature to obtain a disentangled representation without requiring annotations. and we observe a usable level of sparsity and a strong level of alignment with human perception. In applications where there is a need for stronger alignment, we believe that a few hours of human annotation would be sufficient for guiding the representation. We have included this information in a new Section called “Limitations and Future Work”

---

### Author Response · Authors · 2020-11-25
**General Response to the Reviewers (2/2)**

**Could you include discussion, limitations, and future work (R2, R3) answering the following questions? Q1) What happens if your encoder-decoder is biased (R2)? Q2) What happens when your encoder-decoder is not able to produce disentangled representations (it seems often the case on real and complex datasets) (R2)? Q3) How to calibrate the many hyperparameters (to train the encoder-decoder and to weight your loss functions) (R2)? Q4) Why were other data types not treated (R3)?**

Thank you for the suggestion! We have added a new section titled “Limitations And Future Work” at the end of the paper. It addresses each of these questions.

Q1) If the encoder-decoder is biased one could expect the counterfactuals to be biased as well. However, as shown in the new Synbols experiment (results in Figure 2c), our VAE can be trained using a different dataset and still find non-trivial explanations.  In that way, it could be trained on a larger unlabeled dataset to overcome possible biases caused by the lack of annotated data.

Q2) In the case where the generative model would be heavily entangled it would fail to produce explanations with a sparse amount of features.  However, our approach can still tolerate a small amount of entanglement, yielding a small decrease in interpretability. We expect that progress in identifiability (Locatello et al. 2020; Khemakhem et al. 2020) will increase the quality of representations. With a perfectly disentangled model, our approach could still miss some explanations or biases. E.g., with the spectral clustering of the Fisher, we group latent variables and only produce a single explanation per group in order to present explanations that are conceptually different. This may leave behind some important explanations, but the user can simply increase the number of clusters or the number of explanations per cluster for a more in-depth analysis.  In applications where there is a need for stronger alignment, we believe that a few hours of human annotation would be sufficient for guiding the representation

Q3) Finding the best hyperparameters for VAEs is an open problem. In our submission, we chose the VAE with the lowest ELBO. As for the rest of the hyperparameters, we report a distribution in Figure 2 to provide additional insights regarding the limitations and sensitivity of our method.

Q4) As for other data types, we were motivated by the approach presented in the ICLR2020 spotlight paper Progressive Exaggeration (Singla et al. 2020), but our work could be extended to other domains. For example, our method could be applied to find non-trivial explanations on tabular data by directly optimizing the observed features instead of the latent factors of the VAE. However, further work would be needed to adapt the DiVE loss functions to produce perturbations on discrete and categorical variables.

Khemakhem, Ilyes, et al. "Variational autoencoders and nonlinear ica: A unifying framework." International Conference on Artificial Intelligence and Statistics. PMLR, 2020.
Locatello, Francesco, et al. "Weakly-Supervised Disentanglement Without Compromises." arXiv preprint arXiv:2002.02886 (2020).

**Could you restructure the related work (R2), and include missing references (R3)?**

Thanks! We have restructured the related work and extended it to include these additional references you suggested.

---

### Author Response · Authors · 2020-11-25
**General Response to the Reviewers (1/2)**

We thank the reviewers for their thoughtful feedback. We are encouraged by their positive comments: that they found our method interesting (R1, R2, R3), that we identified relevant issues (R2), that it is a sound formulation (R3), and that the experiments aid the reader’s understanding of our contributions (R4). We are glad R2 really liked our proposed method and found that the improvements over the state-of-the-art are significant. We are also pleased the reviewers found our work well contextualized, easy to read, easy to understand (R2), and that the bias detection case study is effective and well presented (R3).

We have updated and improved the submission by taking into account all the reviewers’ comments. In the updated submission, we have highlighted all the changes in blue. Below, we first address questions raised by multiple reviewers and then answer specific questions as individual replies to each reviewer. Any references to Tables and Figures refer to the updated submission unless otherwise stated.

**(R1, R3) Could you clarify/formally define the definition of “trivial” and “valuable” and make it consistent across sections?**

Yes, we have updated the definition of “valuable” to “explanations that are valid, proximal, sparse, and non-trivial”. The first 3 properties are defined as in previous literature (Mothilal et al. 2020) and we introduce non-trivial as “not changing the main attribute the classifier has been trained to identify”. For instance, an explanation that suggests to increase the "smile" attribute of a "smile" classifier for an already-smiling face is trivial (and it does not explain why a misclassification occurred). We have consistently changed this in the whole paper.

**(R1, R2) How is data split for training and evaluation?**

For fair comparison, we used the same splits and evaluation protocol introduced in Progressive Exaggeration (Singla et al. 2020). In particular, CelebA is partitioned into a training and validation set. The generative model is trained on the training set, and hyper-parameter search is performed on the validation set. Likewise, the explainer hyperparameters are found using the training set by random search and results are reported on the validation set. In Figure 2, however, we report the mean and probability density estimation for the whole hyperparameter space. We have clarified this point in Section 4 of the update submission.

**Could you produce explanations on a different dataset (R1) with simpler features such as MNIST or CIFAR (R3)?**

Yes. We have included out-of-distribution experiments using the Synbols benchmark  (Lacoste et al., 2020), which is more challenging than MNIST. It is also different from other small benchmarks such as CIFAR since it includes attributes such as “font” that can help to evaluate the explanations’ sparsity and non-triviality. In this experiment, we train on a subset of the characters in the alphabet and generate explanations for another disjoint subset of characters. Results are reported in Figure 2c, where we observe that DiVE (our method) is successful at finding non-trivial explanations in the OOD scenario.

Lacoste, Alexandre, et al. "Synbols: Probing learning algorithms with synthetic datasets." Advances in Neural Information Processing Systems 33 (2020).

**It is difficult to assess the quality of the generated explanations without human evaluations (R4), do you think we can trust the oracle (R2)?**

We cannot completely trust the oracle. However, we know it was pre-trained on a dataset larger than CelebA for a different task and thus it may not share the same biases as our model. Also the performance of the oracle on the validation set is higher than the performance of the classifier being explained. Another reason for using the oracle is that it had already been tested in progressive exaggeration (PE), Section 4.4.

Note that neural networks are also used as an approximation to unbiased estimators in other domains such as image generation (FID, Inception Score). Nonetheless, in the updated version of the manuscript we have included a human evaluation (Appendix F), showing that the humans and the oracle coincide 63% of the time for DiVE$_{FisherSpectral}$ while only 50% for xGEM. This indicates that DiVE is more successful at finding counterfactuals that do not change the main attribute being classified by the ML model. In addition, we provide the correlation coefficients between the human and the oracle's predictions. A statistical inference test reveals a significant correlation (p-value=0.02).

(1/2)

---

### Decision · Program_Chairs · 2021-01-07
**Final Decision**

**Decision:**

Reject

**Comment:**

The authors propose to use counterfactual (a.k.a as contrastive, as they do not account for causal mechanism) explanations  to explain the errors of an already trained predictive model with images as input data. To this end the authors rely on the manipulation of the latent space of  a VAE with disentangled representations. In general the idea is simple (and based on approaches from prior works) and extends work on counterfactual explanations which have been broadly studied in other domains like decision making where the input data have often semantic meaning (in contrast with the pixel of an image).  While the technical contribution is quite limited, I believe that the general approach of the paper is interesting.

However, even after the rebuttal and reading the updated version, it is still unclear what exactly means key concepts for the paper like trivial and actionable, and more importantly how to use the proposed approach in practice, beyond checking/correcting for potential gender bias in  the data (given that you have access to gender information). In particular, it is not clear how you measure "non-triviality" for the predictive task when you do not have additional knowledge (like the gender) or when the disentangled latent representation do not correspond to semantical features (which as far as I understand they do not need to).  Similarly, while actionability and diversity have been broadly discussed in the decision making domain, it is again not clear what an actionable feature means here to me. Is it just that you can perturb the latent space?

Furthermore, I believe it is worth exploring the connection to approaches for adversarial examples. As it has already discussed in the literature, in terms of formulation,  counterfactual explanations resemble the problem of adversarial examples, but it seems substantially different semantically. At times when reading the paper, it feels that it is indeed more related to adversarial examples than to counterfactual explanations, as the explainability part seems quite superficial. Thus, I would encourage the authors to better position their paper.

In summary, I believe that the paper requires further work before being ready for publication. In particular, the paper would significantly from: i) a better positioning of paper with respect to the literature; ii) formally introduce  key concepts like actionability, diversity and triviality, explaining what they mean in this context, and how to measure them; and more importantly, iii)  explaining how the proposed approach (which by the way involves training a generative model)  can be used in general to 'understand' a model. On a final note, I believe that the paper would benefit from from bringing back to the main body of the paper the experiments that were moved to the appendix during the rebuttal.